# Terrestrial land cover shapes fish diversity in a major subtropical river catchment

Heng Zhang [1,2], Rosetta C. Blackman [1,2], Reinhard Furrer [3,4], Maslin Osathanunkul [5], Jeanine Brantschen [1,2], Cristina Di Muri [6], Lynsey R. Harper [7], Bernd Hänfling [8], Pascal A. Niklaus [1], Loïc Pellissier [9,10], Michael E. Schaepman [11], Shuo Zong [9,10] & Florian Altermatt [1,2] ✉

Freshwater biodiversity is critically affected by human modifications of terrestrial land use and land cover (LULC). Yet, knowledge of the spatial extent and magnitude of LULC-aquatic biodiversity linkages is still surprisingly limited, impeding the implementation of optimal management strategies. Here, we compiled fish diversity data using environmental DNA (eDNA) sampling across a 160,000-km$^2$ subtropical river catchment in Thailand characterized by exceptional biodiversity yet intense anthropogenic alterations, and attributed fish species richness and community composition to contemporary terrestrial LULC across the catchment. We estimated a spatial range of LULC effects extending up to about 20 km upstream from sampling sites, and explained nearly 60% of the variance in the observed species richness, associated with major LULC categories including croplands, forest, and urban areas. We find that integrating both the spatial range and magnitude of LULC effects is needed to accurately predict fish species richness. Further, projected LULC changes showcase future gains and losses of fish species richness across the river network and offer a scalable basis for riverine biodiversity conservation and land management, allowing for potential mitigation of biodiversity loss in highly diverse yet data-deficient tropical to sub-tropical riverine habitats.

Globally, human activities severely threaten biodiversity, challenging sustainable development goals proposed by the Intergovernmental Science-Policy Platform on Biodiversity and Ecosystem Services[1,2]. Biodiversity is declining, with losses of genetic, taxonomic, and functional diversity observed across all ecosystems, putting up to one million species at risk of extinction[1,3]. Freshwater ecosystems harbor an exceptionally high diversity of taxa[4], supporting 43% of all known fish species, with many drainage basins containing taxonomically and functionally unique fish assemblages[4,5]. Yet, many of these fish species are threatened worldwide, especially in riverine systems[6]. The decline of fish species richness has been widely attributed to major global change impacts in respective rivers, including modifications of river connectivity due to hydroelectric dams, warming and oxygen depletion of the water, overloading of nutrients,

chemical pollution, or direct exploitation by fishery[4,7]. Among all the factors, terrestrial land use and land cover (LULC) are recognized as a strong determinant for fish diversity and community distribution in most parts of the world[8,9], with potential impact within a certain distance downstream. However, due to the limited spatial understanding of LULC-fish species associations, attributing current and predicting future effects of terrestrial LULC changes on fish diversity remain challenging, particularly in highly biodiverse yet data-deficient regions[10]. These impede concrete and enforceable approaches to conservation management.

Riverine systems and surrounding terrestrial ecosystems are tightly interconnected at the catchment scale, resulting in cross-ecosystem linkages of resource and pollution flows[11]. LULC and its change thus impact river biodiversity through this terrestrial-aquatic coupling[12,13]. Despite many

---

[1]Department of Evolutionary Biology and Environmental Studies, University of Zurich, Zürich, Switzerland. [2]Department of Aquatic Ecology, Eawag, Swiss Federal Institute of Aquatic Science and Technology, Dübendorf, Switzerland. [3]Department of Mathematics, University of Zurich, Zürich, Switzerland. [4]Institute of Computational Science, University of Zurich, Zürich, Switzerland. [5]Department of Biology, Faculty of Science, Chiang Mai University, Chiang Mai, Thailand. [6]National Research Council (CNR), Research Institute on Terrestrial Ecosystems (IRET), Lecce, Italy. [7]The Freshwater Biological Association, The Hedley Wing, YMCA North Campus, Lakeside, Newby Bridge, Cumbria, UK. [8]Institute for Biodiversity and Freshwater Conservation, University of the Highlands and Islands, Inverness, UK. [9]Ecosystems and Landscape Evolution, Institute of Terrestrial Ecosystems, Department of Environmental Systems Science, ETH Zürich, Zürich, Switzerland. [10]Unit of Land Change Science, Swiss Federal Research Institute for Forest, Snow and Landscape Research (WSL), Birmensdorf, Switzerland. [11]Remote Sensing Laboratories, Department of Geography, University of Zurich, Zürich, Switzerland. ✉e-mail: Florian.Altermatt@ieu.uzh.ch

efforts, it remains difficult to estimate the spatial extent and magnitude of terrestrial LULC effects on riverine fish communities, and the estimates are subject to considerable uncertainties[14]. For instance, croplands and urban areas, two typical LULC types associated with human activities, have strong effects on fish species richness in rivers, yet they are individually assessed at local scales with high uncertainties and cannot be used to accurately predict fish diversity[15–20]. Reported spatial ranges of the terrestrial LULC effects vary from dozens of meters to hundreds of kilometers downstream, yet the corresponding studies commonly focus on a few species and/or LULC types[16,21]. Surprisingly, the fragmented mosaic structure and dynamic nature of LULC are regularly overlooked in assessments of terrestrial LULC impacts on highly biodiverse riverine systems. Combined, this leads to an insufficient understanding of terrestrial LULC effects on fish diversity.

Large subtropical river catchments are global biodiversity hotspots, harboring among others a fascinating diversity of fish species[22]. One of these is the Chao Phraya River catchment in Thailand, holding many native yet threatened species such as the Siamese giant carp (*Catlocarpio siamensis*) or the endemic redtail shark minnow (*Epalzeorhynchos bicolor*). This biodiversity, however, is threatened by anthropogenic changes, including the intensification and expansion of agricultural activities over the past centuries. Croplands today occupy almost all the plains, and currently even expand into hilly and mountainous regions, therefore reducing natural ecosystems such as forests and shrublands. Urban areas have also expanded

rapidly in recent decades, with a four-fold increase in area from 1992 to 2016[23]. This centuries-long and currently accelerating impact is predicted to intensify even further in the coming decades, leading to a loss of forest cover and threatening biodiversity in this region[24].

Here, we developed a spatially explicit model that incorporates LULC maps and fish diversity assessments from environmental DNA (eDNA) sampling to quantify terrestrial LULC effects on riverine fish distributions. We then applied this modeling framework to the Chao Phraya catchment and projected the pattern of fish species richness and distributions. Specifically, we provide a quantitative assessment of the spatial range of the effects of major terrestrial LULC types on fish species richness, and quantify the magnitude of these effects across the river catchment. Further, we project past and future fish diversity using historical and predicted future LULC data, identifying river habitats of fish species of conservation concern.

## Results

### A spatially explicit model

Fish communities along the major river channels in the Chao Phraya catchment were sampled using river water eDNA collected under base-flow conditions (Fig. 1). We used fish eDNA sampling because of its high congruence with traditional sampling methods[25]. The detailed procedures are described in the "Materials and Methods" section and in Blackman et al.[26]. In

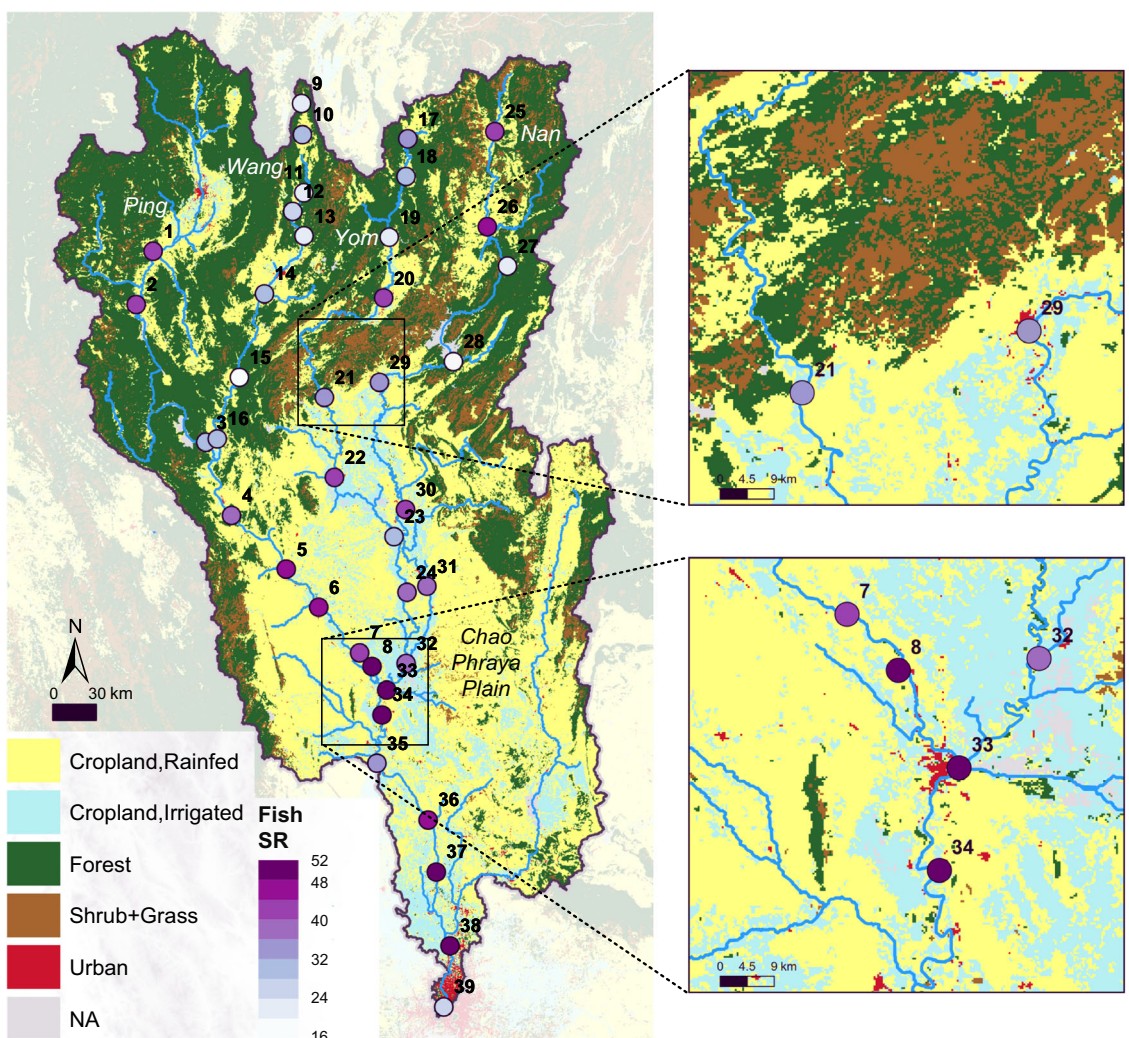

**Fig. 1 | Riverine fish species richness (SR) and terrestrial land use and land cover (LULC) in the 160,000 km² Chao Phraya catchment in Thailand.** Fish species richness of 39 sites derived from environmental DNA (eDNA) in the major river channel is shown in purple dots, representatively covering the whole catchment. Zoomed-in figures depict how the urban LULC type, with overall little area coverage, is especially predominant close to the major river channels.

**Table 1 | Estimations of FishDiv-LULC model parameters, including terrestrial land use and land cover (LULC) effect values on riverine fish species richness and the spatial range *r* by flow distance**

|  | a | Crop (R) | Crop (I) | Forest | S. + G. | Urban | b | r (km) | adj.R² | -2l |
|---|---|---|---|---|---|---|---|---|---|---|
| Value | 20.585 | 1.438 | −0.238 | −2.163 | −4.857 | −3.684 | 3.550 | 19 | 0.587 | 250.613 |
| p | — | 0.054 | 0.852 | 0.055 | 0.631 | 0.019 | 0.003 | <0.001 | — | — |

The magnitude of terrestrial LULC effect of rainfed cropland, irrigated cropland, forest, shrub- and grassland (S. + G.), and urban areas, respectively, is provided. *l* is the log-likelihood function for optimization (see Supplementary Text). The significance of parameters is determined by a likelihood-ratio test (see "Materials and Methods" section and Supplementary Text). The uncertainty estimation is indicated in Table S8. We found a significant relative positive effect of rainfed cropland on fish species richness, but significant relative negative values of forest and urban areas across the Chao Phraya catchment.

brief, we sampled water from 39 sites in 2016, which was then filtered, DNA extracted, and thereafter sequenced using two pairs of 12S primers (Kelly primers for vertebrates and MiFish primers for fish)[27,28]. From these two primer sets, we obtained in total 5,825,212 and 4,927,576 reads, respectively, which were merged by taking maximum read counts and matched with a total of 108 fish taxa (mostly at the species level, and subsequently referred to as fish species). At each site, species-level records were converted into presence/absence to calculate species richness. Among these fish species, seven were identified as critically endangered (CR), endangered (EN), vulnerable (VU), or near threatened (NT), according to the Red List of the International Union for Conservation of Nature (IUCN), after having removed alien and invasive species (Table S1). Across the catchment, fish species richness ranged from 13 to 52 (34.5 on average), with strong variation across the river network (see also results reported in Blackman et al.[26]). In general, fish species richness was high in the lower reaches of the Chao Phraya (Fig. 1), covering most of the plain area. In contrast, upstream reaches, generally in hilly or mountainous regions, showed low species richness but highly distinct communities among tributaries[26]. Nevertheless, sites in hilly or mountainous regions near croplands showed relatively high species richness (mean = 34.9) compared to sites away from croplands (mean = 20.9) (Fig. S1).

Terrestrial LULC was quantified using a 300 m-resolution land cover map (reference year 2016; European Space Agency Climate Change Initiative (ESA CCI)[23]). We recoded the original 36 LULC classes into the five predominant and distinct LULC types: rainfed cropland (44% cover), irrigated cropland (12% cover), forest (36% cover), shrub- and grassland (7% cover), and urban (1% cover) (Fig. 1; Table S2). Croplands were the dominant LULC type, covering 56% of the catchment area, and were mainly found in the plains and along the rivers. Forest was predominantly found in the mountainous region at a higher elevation and included a combination of broad-leaved evergreen and deciduous forests. Urban areas, though only occupying ~1% of the catchment area, were commonly found in the direct vicinity of the major river channels. They thus had a high potential to influence riverine fish species composition.

For each eDNA sampling site, we produced a map of flow distances based on the three-arc-second resolution HydroSHEDS (version 1) flow direction map[29], i.e., we determined, for each pixel, the distance of the water flow path that connected it to the sampling site (see "Materials and Methods" section). These flow distance maps were then resampled to match the ESA CCI land cover map. Overall, the maximal flow distances to the sampling sites ranged from 31 to 1076 km upstream (Fig. S2). In addition, we extracted river discharge, a parameter relevant to the fish diversity gradient and river characteristics, from the HydroSHEDS database as a predictor of baseline fish species richness (Fig. S3)[30,31].

Combining riverine fish diversity, LULC, and catchment data, we created a spatially explicit model (hereafter referred to as the FishDiv-LULC model) by considering the spatial range and magnitude of effects from terrestrial LULC types on fish species richness. This model linked observed fish species richness to the terrestrial LULC effects integrated upstream along the flow path, and explained 58.7% (adjusted $R^2$) of the variance in the observed fish species richness (Table 1, see "Materials and Methods" section). LULC-only and river discharge-only effects explained 21.7% and 9.0%, respectively, of the total variance in fish species richness, demonstrating significant terrestrial LULC effects on fish diversity in this

catchment (Fig. S4). The estimated spatial range of terrestrial LULC effects on local fish species richness was 19 km (90% CI: 11–34 km) upstream from sampling sites, suggesting a scale at which terrestrial LULC effects modulate fish species richness in the river. Rainfed cropland had a significant relative positive effect on fish species richness, whereas forest and urban areas showed relative negative values (Table 1; see "Materials and Methods" section). It is important to emphasize that these relative values (positive or negative) were differences from the baseline estimation across the river catchment. In contrast, rainfed cropland had a much higher positive impact on fish species richness, thereby increasing baseline fish species richness across the catchment, resulting in relatively negative values in forest and urban areas.

## Validation, robustness, and estimation of uncertainties

We assessed the high robustness of our findings by comparing estimation results in two spatially separated sub-regions, splitting the fish sampling dataset in half. We separated 19 out of the 39 sites belonging to Northern Thailand in the hillier and more mountainous region with an elevation above 100 m; the remaining 20 sites were in Central Thailand, a plain-dominated region with an elevation below 100 m (Fig. S5, see "Materials and Methods" section). For both datasets, we independently found similar approximate positive terrestrial LULC effects from rainfed cropland, and approximate negative values from forest and urban areas, though the estimated spatial range of these effects differed between regions (Table S3). This indicated that the estimated LULC-fish species richness association was not driven by spatial clustering of the mountain and lowland regions, but by the LULC types themselves. It further indicated that even a smaller sampling effort (~20 sites) was sufficient to approximate LULC effects. We further corroborated the observed terrestrial LULC effects in comparison to a null model. Specifically, we applied a neutral meta-community (NMC) model simulating fish species richness considering climate, fish habitat capacity, speciation, extinction, migration, and river network structure[31], yet excluding any possible LULC effect (see "Materials and Methods" section). Our results showed that the FishDiv-LULC model (adjusted $R^2 = 0.587$) explained a higher amount of variance than the NMC model (adjusted $R^2 = 0.255$) and better captured fish species richness patterns in this subtropical region (Fig. S6 & Table S4), demonstrating strong terrestrial LULC effects on riverine fish species richness.

## Terrestrial LULC effects and fish traits

Mechanistically, we found that terrestrial LULC drove fish species distributions through fish traits, which could be ascribed to the alteration of riverine habitats[32,33]. To this end, we determined the associated terrestrial LULC type for each fish species through species-level modeling, then linked the determined LULC type to fish morphological traits extracted from the FISHMORPH database[34], related to fish life cycle, ecology, and functional roles (see "Materials and Methods" section)[32,35]. We analyzed envelopes of LULC type-associated fishes in trait space (in ordination space generated using a principal component analysis), and found distinct envelope shapes and trait values among different LULC types (Fig. 2, Table S5). Fish species associated with rainfed cropland had a relatively high mean and range of high body elongation (BEl); irrigated cropland-associated species showed relatively high maximum body

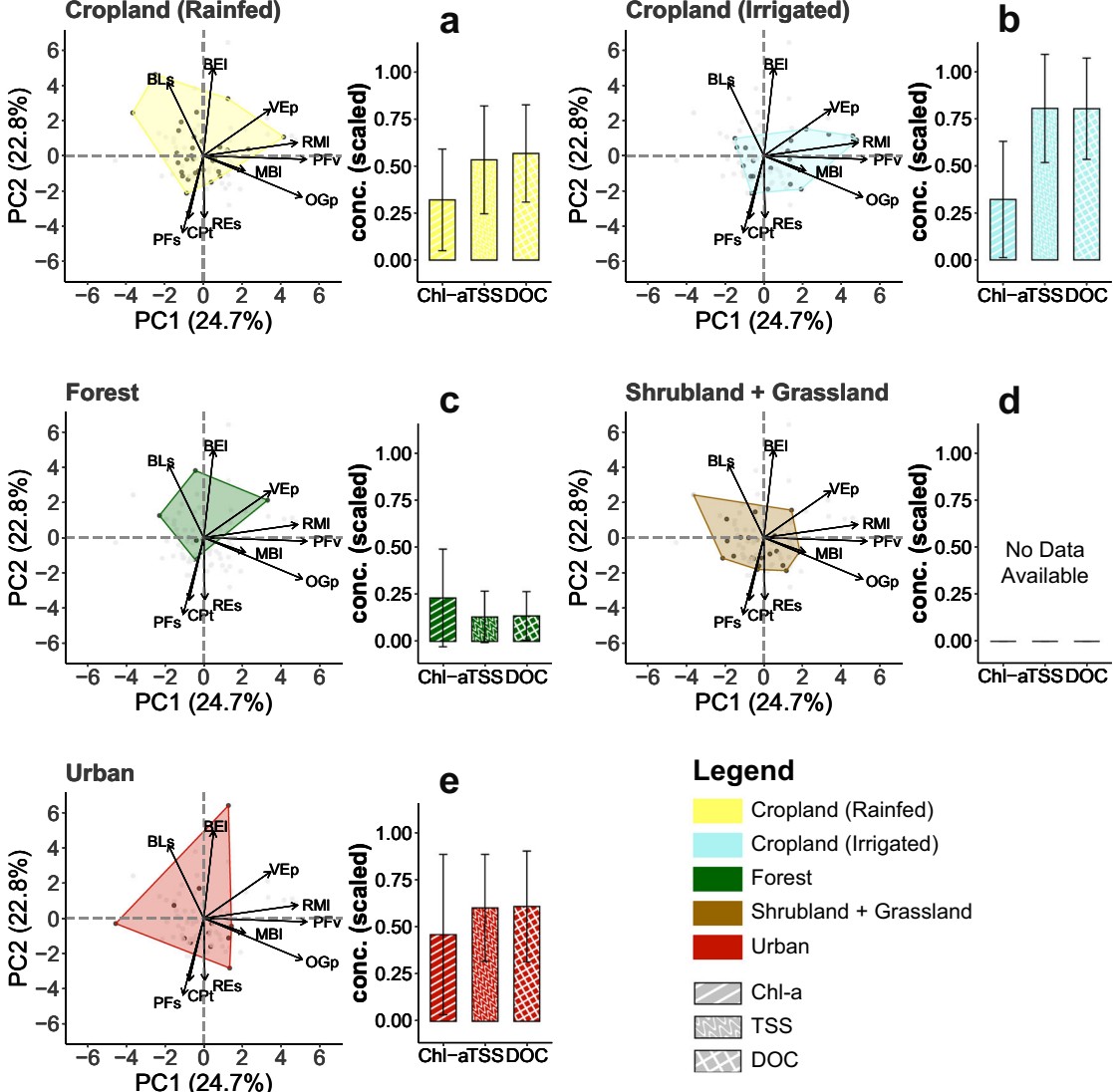

**Fig. 2 | Fish species traits in relation to terrestrial LULC types in the Chao Phraya catchment.** Fish species functional traits and water properties including chlorophyll-a (Chl-a), total suspended solids (TSS), and dissolved organic carbon (DOC) in relation to rainfed cropland (**a**), irrigated cropland (**b**), forest (**c**), shrub- and grassland (**d**), and urban (**e**) land use and land cover (LULC) types, respectively, across the Chao Phraya catchment. The associated LULC for fish species was determined by the highest terrestrial LULC effect ($V_k$) value from species-level (presence/absence) modeling. The trait space envelopes of LULC-associated fishes were created based on the FISHMORPH database with a principal component analysis of the functional trait space of fish species. Maximum body length (MBl), body elongation (BEl), vertical eye position (VEp), relative eye size (REs), oral gape position (OGp), relative maxillary length (RMl), body lateral shape (BLs), pectoral fin vertical position (PFv), pectoral fin size (PFs), and caudal peduncle throttling (CPt) were used to create fish trait space. Distinct fish trait space envelopes were observed among the five LULC types. Water properties were estimated from Sentinel-2 data based on the major river channel pixels with a river width >60 m, and were min-max scaled for plotting. There was no data on water properties for shrub- and grassland types. The error bars indicate the standard deviation. These figures illustrate that rivers in forested areas tend to have lower Chl-a, TSS, and DOC values, thereby less resource or nutrient subsidies compared with cropland and urban areas.

length (MBl) and oral gape position (OGp), relative maxillary length (RMl), pertaining to overall high trophic levels which coincided with high nutrient loadings from surroundings; forest-associated species had comparatively small MBl and caudal peduncle throttling (CPt), suggesting small body size yet agility in swimming; urban-associated species had a high range of MBl, high value of body elongation (BEl) and CPt, and a large trait envelope area, indicating various strategies to adapt to high environmental and hydrological disturbances (Fig. 2, Table S5). Additionally, fish traits themselves were not significantly correlated with river network characteristics, such as river discharge, thus excluding a direct fish trait-river network linkage (Fig. S7). As a consequence, these results implied a strong linkage of terrestrial LULC on individual fish species distributions through fish traits and explained the formation of fish species richness patterns.

## Fish diversity projections with LULC changes

Conservation of riverine biodiversity still lacks effective methods to assess the conservation potential of adjacent terrestrial land and the latter's quantitative impacts on riverine biodiversity. We show how the spatially explicit approach allows for projections of future richness and communities of riverine fish diversity, using minimal information accessible through global LULC products and river water eDNA sampling. To begin with, we traced the flow path within the effective distance ($r$) and determined if it had an effect on riverine fish species for each terrestrial pixel (Fig. S8 for schematic diagram). Then, based on this spatial information, we calculated the integral of each respective pixel's LULC effect along the river channel to produce a map of terrestrial LULC effects on riverine fish species richness (Fig. 3a, see "Materials and Methods" section). This map (Fig. 3a) can be understood as the change of fish species richness per kilometer in the river

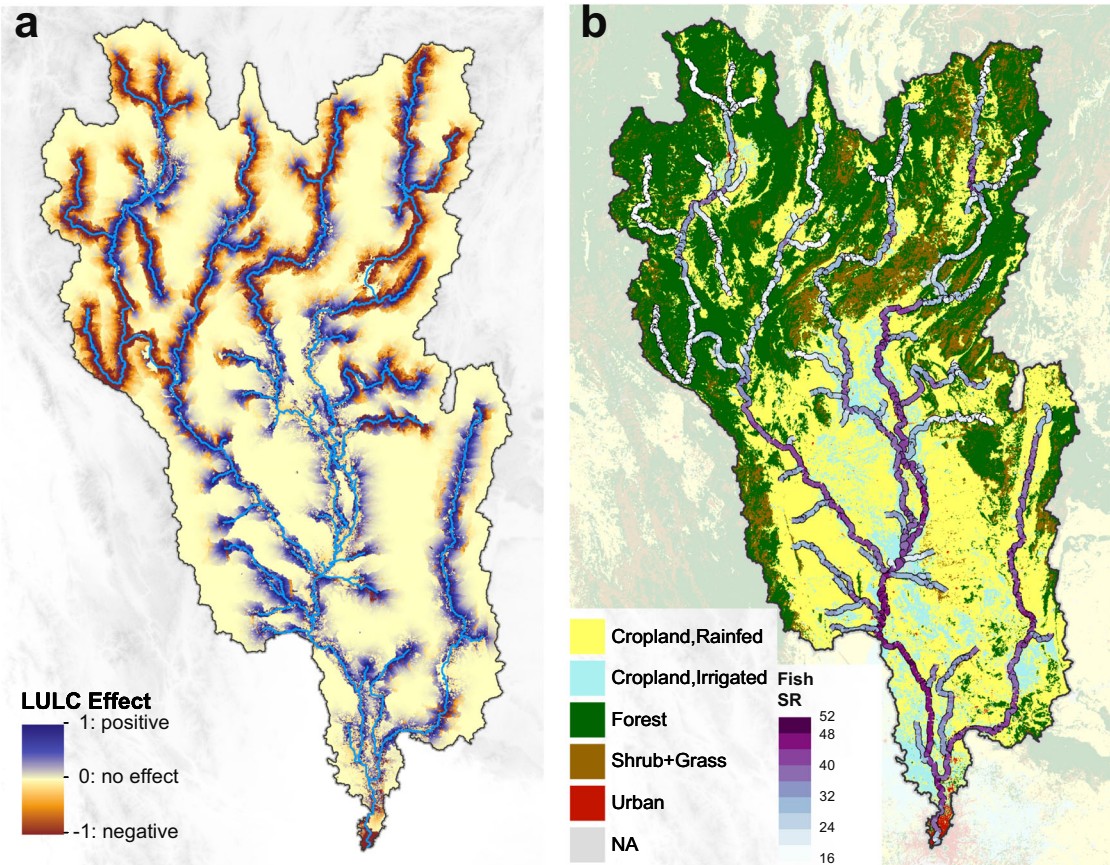

**Fig. 3 | Maps of terrestrial LULC effects and projected riverine fish diversity pattern in the Chao Phraya catchment. a** Map of terrestrial LULC effects ($E_{LULCj}$) on riverine fish species richness in the major river channels of the Chao Phraya catchment. The value is the change of fish species richness per kilometer in the river due to the LULC effect from terrestrial land (unit: number of species/km).

**b** Projected pattern of riverine fish diversity (species richness, SR) in the major river channels of the Chao Phraya catchment. The LULC map is embedded as the background layer. The projection shows a high consistency with eDNA-derived fish diversity sampling in Fig. 1.

due to LULC effects from the terrestrial land. We then projected fish species richness in the major river channels, where regions rich in fish species were mostly observed in the plains and some hilly and mountainous regions close to the croplands, successfully capturing the spatial variation of fish species richness (Fig. 3b).

Anthropogenic LULC changes continue to intensify worldwide[36]. To explore potential impacts of such LULC changes, we used past and modeled future LULC maps to retrospect and forecast riverine fish species richness patterns. To isolate terrestrial LULC effects, we assumed that climate, flow discharge, and river network remained constant. For the past, the change of 24 years of historically observed LULC (1992–2016) was evaluated by adopting the ESA CCI land cover map of 1992[23] (Fig. 4a); for the future, 34 years of modeled LULC changes under future scenarios (2016–2050) were assessed according to the products of GLOBIO4 scenario data[37]. Land use maps of 2050 under shared socio-economic pathway 1 and representative concentration pathway 2.6 (SSP1 RCP2.6), SSP3 RCP6.0, and SSP5 RCP8.5 scenarios were used for forecasting future fish diversity (Fig. 4d for SSP5 RCP8.5, Fig. S9 for other scenarios). According to these LULC maps, urban areas increased by 295% from 1992 to 2016, and were forecasted to increase by another 73% from 2016 to 2050 (under the SSP5 RCP8.5 scenario), mostly along the major river channels. Moreover, predicted LULC changes in the future contained a conversion of forest, shrub, and grassland to croplands, leading to a 25% increase in croplands and a 26% decrease (under the SSP5 RCP8.5 scenario) in forest, mostly in the hilly and mountainous regions. Based on these scenarios, we predicted fish species richness for 1992 (past) and for 2050 (future; under the three

scenarios) (see "Materials and Methods" section), and produced maps of percentage of fish species richness changes (Fig. 4b for the past, Fig. 4e for the future under SSP5 RCP8.5, Fig. S9 for other scenarios). From 1992 to 2016, we observed only slight changes in fish species richness (Fig. 4b), partly due to local urbanization and LULC change from shrub- and grassland to forest (a 21% decrease in shrub- and grassland); yet, from 2016 to 2050, we forecast a remarkable increase of fish species richness in the hilly and mountainous region under three scenarios (7.9—14.3%) because of a strong expansion of croplands by 29—54% in that region (Fig. 3e under SSP5 RCP8.5, Fig. S9 for other scenarios). However, these predictions about species richness did not reflect impacts of terrestrial LULC on less-common and endangered fish species, but mostly on already benefited common or generalist fish species[38].

We therefore modified our FishDiv-LULC model into a species-level model, estimating terrestrial LULC effects on individual fish species and predicting fish species distributions under LULC change scenarios (see "Materials and Methods" section). Briefly, we repeated the future projections, this time using the species-level model and focusing on the subset of seven fish species that were of conservation concern (SPCC) (Table S1, Fig. 4c, f under SSP5 RCP8.5, Fig. S9 for other scenarios; see "Materials and Methods" section). Not unexpectedly, we found that the patterns of change in SPCC richness diverged from the patterns of change in overall fish species richness, with a high inconsistency in the hilly and mountainous regions. This indicated that the major river channels with the most intense LULC change from natural habitats to croplands tended to lose SPCCs, emphasizing the necessity of land management regulations to mitigate such effects.

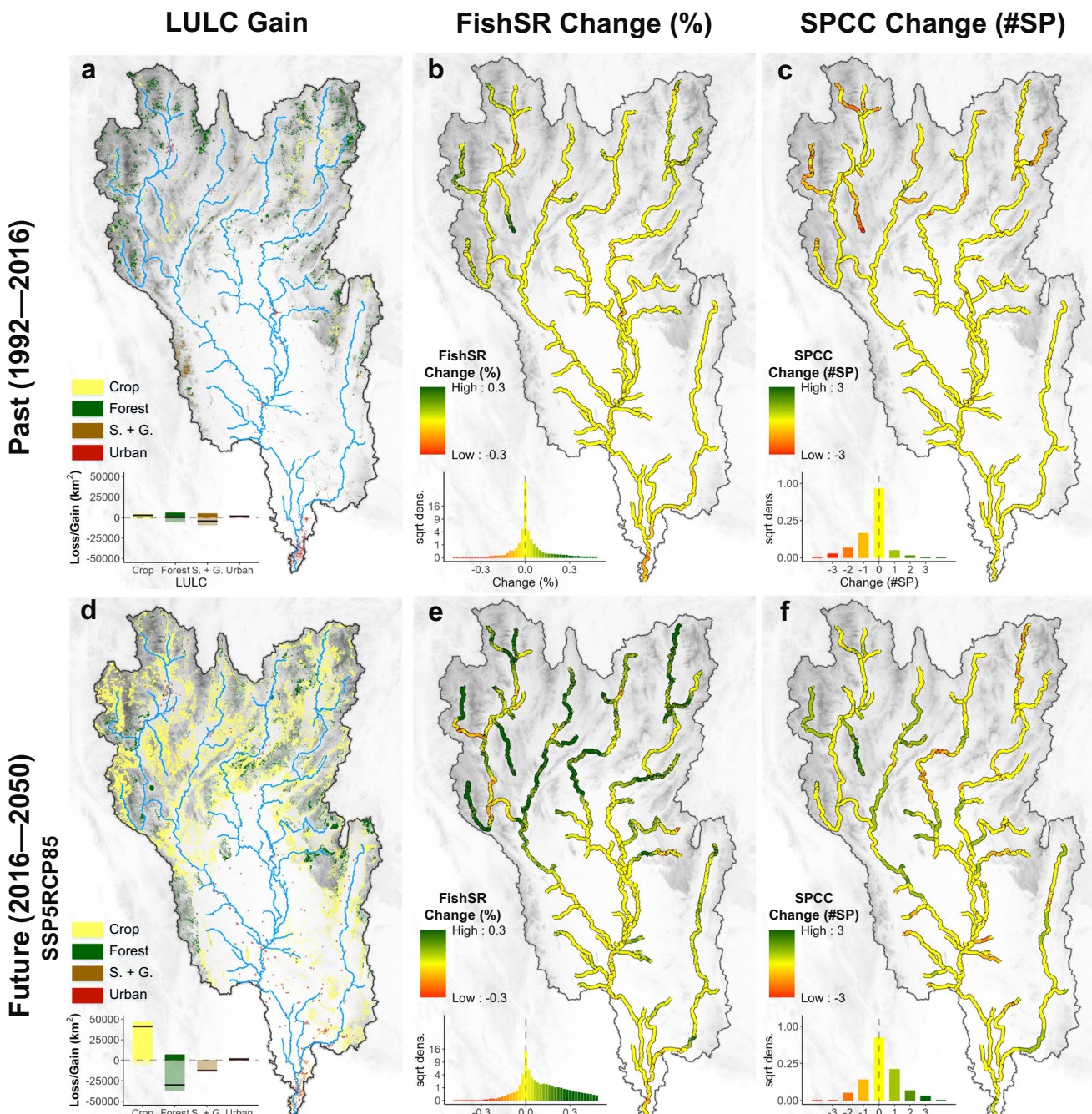

**Fig. 4 | Predicted diversity changes of overall fish species richness (SR) and richness of fish species of conservation concern (SPCC) due to LULC changes in the periods of 1992–2016 (past) and 2016–2050 (future).** Future prediction of LULC was adopted from GLIOBIO4 data, and the list of seven SPCC was derived from the Red List of the International Union for Conservation of Nature (IUCN) with alien or invasive fish species removed (Table S1). We observed croplands and urban areas expansion over the past (**a**) and future (**d**) periods, resulting in a reduction of natural habitats such as forest, shrub, and grassland, especially in the mountainous region in the period of 2016–2050. Projected percentages of overall fish species change between 1992 and 2016 (**b**), and between 2016 and 2050 (**e**). Projected numbers of SPCC change between 1992 and 2016 (**c**), and between 2016 and 2050 (**f**). These predictions demonstrate different trends of fish species richness change between overall fish species richness and SPCC richness. Please refer to Fig. S8 for other scenarios SSP1RCP26, SSP3RCP60.

## Discussion

We demonstrate how a spatially explicit model integrating terrestrial LULC and eDNA-based fish diversity allows for attributing and forecasting of terrestrial LULC effects on riverine fish species richness in a large subtropical catchment. Specifically, for overall fish species richness in the Chao Phraya catchment in Thailand, we found a relative positive terrestrial LULC effect from rainfed cropland, most likely caused by high resource and nutrient subsidies, but estimated relative negative LULC values of forest and

urban areas, with a maximal distance of up to 19 km upstream from sampling sites. By analyzing fish traits in relation to specific terrestrial LULC types, we derived characteristic LULC-fish trait linkages and thereby explained the possible formation of fish species richness patterns. Furthermore, forecasts over future LULC change scenarios indicated strong human impacts on future fish diversity patterns, especially in the hilly and mountainous regions where cropland expansion would increase fish species richness. In contrast, fish species of conservation concern would be

negatively influenced by such LULC changes. Our approach can be applied to other biomes worldwide and allows for attributing fish diversity and its changes to major anthropogenic LULC changes.

The relative positive effect of rainfed cropland on fish species richness that we found aligns with previous case studies, which reported increased fish species richness because of agricultural activities and associated nutrient subsidies, for instance, in Northern Europe and Southern Brazil[15,39]. To show how differences in river nutrient availability influence fish species richness in relation to LULC types, we calculated river chlorophyll-a content (Chl-a), total suspended solids (TSS), and dissolved organic carbon (DOC), water properties reflecting nutrient availability and productivity, using Sentinel-2 level 2A data (see "Materials and Methods" section). In this computation, river channels narrower than 60 m were excluded because no remote sensing in-river values were available. We found higher Chl-a, TSS, and DOC values close to croplands compared to forests (Fig. 2). These data evidence that rivers near croplands received high nutrient run-offs from the surrounding terrestrial land, which could subsequently result in increases in algal biomass and food availability. With enlarged resource availability, especially for more generalist and omni- and algivorous fish species, rivers near cropland can consequently harbor more fish species[15]. In addition, agricultural practices in these areas over the centuries may have already pre-selected fish communities to tolerate some terrestrial impacts from crop farming, such as no or low forest cover, and run-offs of nutrients and sediments. Contrastingly, contemporary wastewater and chemical pollution and high anthropogenic disturbances associated with urban areas affect fish assemblage structure and can cause a decrease in fish species richness in rivers[40].

Importantly, the observed relative positive terrestrial LULC effects from rainfed cropland represented an averaged deviation from the baseline species richness estimation across the whole catchment, or in other words, an increasing trend in fish diversity. This baseline estimation, as suggested, included the effect of river characteristics (e.g., river discharge), the species pool, and an averaged terrestrial LULC effect on fish species richness across the catchment. However, such a relatively positive effect would not actually promote highly specialist fish species, which are often of conservation concern (Fig. 4)[35]. Furthermore, we found no significant positive effects of irrigated cropland, which is a more intense form of agriculture and often associated with excessive nutrient loadings and heavy use of pesticides and fertilizers[41], likely with negative effects on fish diversity[42]. Ultimately, our analysis demonstrated a variety of cropland effects on fish species richness, implying the necessity of adequate and sustainable land management and agriculture regulations in this region.

The expansion of cropland may also cause a decrease in the uniqueness among riverine fish assemblages in the catchment. In the previous results, we focused on patterns of fish species richness, arguably the most commonly used metric of biodiversity[31]. However, this metric is insensitive to uniqueness within communities and has intrinsic limitations. Our further predictions over the future LULC change scenarios showed a decrease in the uniqueness of fish species across the catchment. When calculating the Jaccard similarity index of predicted fish assemblages between sites in the mountainous river and sites in the plain river from 2016 to 2050 under the SSP5 RCP8.5 scenario, we found a remarkable increase in similarity of fish assemblages in the future (Fig. S10, see "Materials and Methods" section). In general, natural habitats have good conservation potential for specialist and narrow-ranged fish species. For example, river channels near forests can harbor more rare fish species, essentially contributing to local biodiversity[43]. Nevertheless, due to cropland expansion into the hilly and mountainous region, native and often endangered species associated with natural habitats could be replaced by cropland-associated and/or wide-ranging species, such as *Osphronemus goramy* or *Boesemania microlepis*, causing a loss of uniqueness[38,39]. Consequently, our future predictions do not suggest that fish diversity loss could be mitigated in the future, but imply that specific land management and regulations are needed to alleviate the adverse impacts of LULC on less-common and endangered species.

Our approach has strong potential to be applied to any river catchment, given high cost-efficiency, high congruence with traditional methods, as well as the comparability of results of river eDNA sampling and globally available high-resolution LULC products[25]. Theoretically, eDNA acquired from flowing water reflects the local and cumulative biodiversity information from the sampling site to a certain distance upstream, which may introduce uncertainties (from tens of meters to a few kilometers) in the spatial assessments[44-46]. These uncertainties depend on the degradation rate of eDNA, which is, among others, influenced by the location of the catchment, temperature, and water quality. In our model, the baseline estimation of fish species richness can be changed to other factors determining fish species richness patterns, such as temperature, habitat size, or drainage area, geological and/or historical factors, river water properties, and river structures[30,31]. In the Chao Phraya catchment, we considered river discharge as a highly relevant parameter for water balance, stream order, and fish habitat capacity. We did not include temperature as a parameter because it was mostly homogenous in this region (Fig. S3a). However, other long rivers, such as the Mississippi, Yangtze, or Danube, flow through multiple biomes and have a larger elevation range; therefore, for those, it may be necessary to include climatic factors and/or river water properties in the baseline fish species richness estimation. In addition, fish assemblages in the river may change from season to season due to migration; therefore, our focus on fish diversity patterns from one dry season may not capture all fish species, and–being a static spatial analysis–is thus a conservative approach. Taking seasonal fish eDNA samples would be helpful to further understand how the magnitude of terrestrial LULC effects would change across seasons; yet we here focused on a single timepoint, also acknowledging and representing the data-scarcity for most (sub)tropical river systems. As such, it is a proof-of-principle that even with a single sampling campaign, biodiversity data can be associated and projected to LULC in such hyper-diverse systems. Evaluating seasonal changes of LULC effects could further help to reveal the dominant mechanisms behind the fish diversity-LULC linkage.

Human modifications of terrestrial landscapes are a primary driver of LULC change, and through cross-ecosystem linkages, are continually reshaping riverine biodiversity. Cropland and urban areas, the typical anthropogenic LULC types having pronounced effects on this biodiversity, can be directly governed by legislation and policies through controlling the area and position in the landscape. Therefore, when developing adequate conservation strategies for freshwater ecosystems, we need careful consideration of current and future LULC spatial distributions. The estimated effective distance of associations between fish diversity and terrestrial LULC is a critical piece of information in designing a conservation buffer zone. For example, many Amazonian countries have adopted riparian protection buffers of tens of meters in width; however, this spatial distance is insufficient to protect fish species given human alterations to terrestrial LULC, and such a spatial design needs a careful estimation of the influential range[47,48]. Global biodiversity protection initiatives, such as the 30 by 30 Initiative[49], aiming to manage the specific use of land, should not only consider the total amount of land but also its spatial position and the underlying cross-ecosystem effects at the catchment level. Our approach directly estimates local fish diversity changes under anthropogenic terrestrial LULC alterations, giving both scientists and stakeholders a potent tool in land management and conservation area design.

## Materials and methods

The study was conducted in the Chao Phraya River catchment located in Northern and Central Thailand, covering rivers in both mountainous and plain landscapes. We combined fish diversity data from eDNA sampling in the rivers (elevation ranging from 2 m to 509 m a.s.l.) and a land use and land cover map across the 160,000 km² catchment (Fig. 1).

### Environmental DNA sampling and fish data

Environmental DNA (eDNA) metabarcoding has become a standard approach for aquatic biodiversity sampling and monitoring due to its high

efficiency and significant reduction in fieldwork intensity[50]. A meta-analysis has revealed that there is a very good congruence in gamma diversity between eDNA sampling for fish species and sampling using traditional sampling methods (on average an almost 1:1 match on richness, and a >50% congruence on the specific fish species found by both methods, and an additional ~25% of species found by either eDNA and traditional monitoring only), showing the potential of replacing traditional especially invasive sampling methods[25]. In this research, the fish data were derived from an eDNA metabarcoding study with methodological details published therein[26]. Briefly, eDNA sampling was carried out in 2016 at 39 sites in major river channels, during the dry season under base-flow conditions. At each site, six samples were collected from the left bank, channel center, and right bank, respectively (two replicates each; 234 samples in total). Following on-site filtration (600 mL water in total per sampling site), DNA was extracted using standardized methods, and metabarcoding analyses were carried out using two separate molecular assays based on the mitochondrial 12S region, subsequently referred to as the Kelly primers for vertebrates and the MiFish primers for fish[27,28]. To improve the accuracy of sequence assignment, we created a customized reference library database from GenBank targeting fish species known to occur in the Chao Phraya River catchment, according to OEPP Biodiversity Series Vol. 4 Fishes in Thailand[51] and the Checklist of Freshwater Fishes of Thailand (http://www.siamensis.org/). During bioinformatic processing, sequences were assigned to a total of 108 fish taxa (mostly at species level, so referred to as fish species), with 82 and 93 taxa recovered using the Kelly primers and MiFish primers, respectively. Differences in species communities between the two assays can largely be accounted for by unequal representation of the respective DNA regions in the reference database and differences in species-level resolution[26]. We merged these two datasets by choosing the higher read counts at each site for each fish species and calculated species richness. The list considered included native and naturalized species, part of which were also used in aquaculture. We further matched the detected fish species with the Red List of the International Union for Conservation of Nature (IUCN), having removed any alien or invasive species. In total, seven species were identified as critically endangered (CR), endangered (EN), vulnerable (VU), or near threatened (NT), and were treated as species of conservation concern (SPCC) in our fish data (Table S1).

## Land use and land cover (LULC) data

We used the European Space Agency Climate Change Initiative (ESA CCI) land cover map, with a yearly interval (website: https://www.esa-landcover-cci.org/)[23]. Specifically, we used the 300 m-resolution map from 2016 to temporally match our fish data. Among the 36 classes in the classification system, 22 classes—including croplands, forests, shrublands, grasslands, and urban areas—were observed in the Chao Phraya catchment.

To alleviate uncertainties from rarely observed LULC types, we excluded or merged those LULC types occupying <0.2% of the area. Further, we removed all areas covered by water (lakes, reservoirs, and rivers), so that only terrestrial LULC types were used. We then recoded this LULC map into a five-class-system map, comprised of rainfed cropland, irrigated cropland, forest, shrubland- and grassland, and urban areas (Fig. 1). A detailed recoding table is shown in Table S2.

## River channel and catchment data

To improve spatial modeling performance, we adopted the three-arc-second resolution (~92 m at the equator) HydroSHEDS (version 1) flow direction map to calculate potential catchments for sampling sites[29]. The value of a given pixel in the water flow direction map represents the direction of water flow for eight neighboring pixels (i.e., the eight possible values represent eight directions of water flow). Therefore, for each sampling site, one can track all the pixels in the map and determine if one pixel ultimately flows through the sampling site. The cumulative flow distance from the pixel to the sampling site can also be calculated at the same time. Then, all the pixels that flow through the sampling site are connected as a catchment map for that sampling site.

Following this approach, we produced catchment maps of all sampling sites with flow distances calculated by the haversine formula (see Supplementary Text) to improve the modeling accuracy. Next, the catchment maps with flow distance were resampled to match the LULC data and then were used in the FishDiv-LULC model introduced below as $d_{ij}$ (the flow distance between a catchment pixel $j$ and sampling site $i$). The major river channels (blue lines in Fig. 1) were extracted using a threshold drainage area of ~810 km$^2$ (100,000 pixels). We also extracted the long-time (1970–2000) average river discharge (unit: m$^3$/s) from the HydroSHEDS database, hereafter referred to as river discharge ($Q$), for all sampling sites and major river channel pixels.

In this study, we chose the ESA CCI LULC data with a resolution of 300 m, but not a map with higher spatial resolution, mainly because of the uncertainties in depicting river channels in this 90 m HydroSHEDS map, even though it was one of the most accurate databases available. We noticed that the meandering river channels in the mid- to lower reaches are not absolutely precise, with spatial uncertainties of up to several pixels, which introduced non-negligible uncertainty into the structure of river networks. For these reasons, we manually checked the location of eDNA sampling sites and tried to minimize the uncertainty in catchment calculations.

## Modeling terrestrial LULC effects on fish species richness (FishDiv-LULC model)

We developed a spatially explicit modeling framework to assess terrestrial LULC effects on riverine fish species richness, considering both the spatial range and magnitude of LULC effects[52]. Briefly, we hypothesize that the fish species richness at a sampling site equals the integral of LULC effects within the site's catchment (with a distance decay framework) plus a baseline estimation. The following modeling and computation are carried out at the pixel level because of the raster format of data used in this study. We have also specified units for all the parameters in the model so that one can directly perceive the actual meaning behind.

To begin with, let $k = 1, 2, …, K$ represent the LULC type, and $j = 1, 2, …, N_{ik}$ represent the pixel index of LULC type $k$ in the catchment map of sampling site $i$ ($i = 1, 2, …, M$; $M = 39$ in this case). Overall, then the observed fish species richness at sampling site $i$ ($B_i$) on the left-hand side is assumed to be equal to the sum of effects of different LULC types in site $i$'s catchment ($\sum S_{ik}$, unit: num. species) plus a baseline fish species richness prediction $B_i^0$ and an error $\varepsilon_i$ (Eq. 1).

$$B_i = \sum_{k=1}^{K} S_{ik} + B_i^0 + \varepsilon_i. \tag{1}$$

Because sites closer to the river have higher effects than sites farther away from the river, we introduce a distance decay framework to characterize this inverse relationship for terrestrial LULC effects over the water flow distance. For each site of interest (e.g., site $i$), the effect of a pixel $j$ with LULC type $k$ on the fish species richness can be written as $V_k \cdot A_j \cdot f(d_{ij})$. Whereby $V_k$ is the magnitude of the effect of LULC type $k$ (unit: num. species/km$^2$), $f(d)$ is a distance decay function, and $A_j$ is the area of pixel $j$ (unit: km$^2$) depending on the coordinates and is estimated by the haversine formula (see Supplementary Text).

We compared and evaluated five commonly used distance decay functions and chose the widely used exponential decay function in this case because of its simplicity (see Supplementary Text and Table S6). In the exponential distance decay function, distance is directly the flow distance in the catchment map ($d_{ij}$) from pixel $j$ to site $i$ (Eq. 2).

$$f(d_{ij}) = \frac{3}{r} e^{-\frac{3d_{ij}}{r}}. \tag{2}$$

Here, the parameter $r$ (unit: km) indicates the effective distance at which the magnitude of terrestrial LULC effect has dropped to ~5% of its original value (a common threshold in spatial statistics). Consequently, the

terrestrial LULC effect is explicitly expressed in Eq. (3).

$$S_{ik} = \begin{cases} \frac{3}{r} \cdot V_k \cdot \sum_{j=1}^{N_{ik}} A_j \cdot e^{-\frac{3d_{ij}}{r}}, & \text{if } LULC_j = k, \\ 0, & \text{if } LULC_j \neq k. \end{cases} \quad (3)$$

Note that $\sum_{k=1}^{K} N_{ik}$ is equal to the total number of pixels in the catchment map of site $i$ (i.e., every pixel belongs to one specific LULC type $k$). The baseline prediction $B^0$ (unit: num. species) is expressed using river discharge ($Q$), which best explains fish species richness pattern (Fig. S3, Eq. 4).

$$B^0 = a + b \cdot \ln Q. \quad (4)$$

## Optimization of model parameters

Model parameters were estimated by solving a maximum likelihood problem, given the observed fish species richness $B_i$. Subsequently, we write the above optimization problem explicitly as Eq. (5) with vector-matrix notation.

$$B \sim \mathcal{N}\left(\mu(\theta), \sigma^2 I\right), \;\; \mu(\theta) = a + b \cdot \ln Q + C(r) \cdot V, \;\; \theta = (r, a, b, V) \quad (5)$$

Whereby $C(r)$ is an $M$-by-$K$ matrix with elements $c_{ik} = 3/r \cdot \sum_{j=1}^{N_{ik}} A_j e^{-3d_{ij}/r}$ depending on the distance parameter $r$; $V$ is a $K$-vector of magnitude parameters $V_k$; $B$ is an $M$-vector of observed fish species richness $B_i$; $a$ and $b$ are constants, with $a$ being the intercept in the estimation; $Q$ is an $M$-vector of river discharge values; and $\theta$ is a $K + 3$ dimensional parameter. We assume each element of the error is independent and identically normally distributed (however, for catchments with a small species pool, Poisson or negative binomial distributions may be considered, and it needs a case-by-case analysis).

Then, we estimate $\theta$ by the maximum likelihood approach (see Supplementary Text for detailed method). We also computed adjusted $R^2$.

## Correlation and significance of model parameters

Variance inflation factor (VIF) and paired Pearson's correlation of parameters were calculated under the estimated effective spatial range ($r = 19$ km). The correlation among the estimated terrestrial LULC effects was relatively weak (Table S7 and Fig. S11). The significance of these effects was determined by a likelihood-ratio test (see Supplementary Text).

## Validation, robustness, and estimation of uncertainties

We used leave-one-out cross-validation to assess the uncertainty of our parameter estimates. Specifically, we reserved one sampling site from the fish data for testing, followed by estimating the parameters based on the remaining 38 sites (i.e., the training set). Then, we predicted the fish species richness value on the reserved site and compared the predicted value with the real observation. We repeated the whole process for each site and calculated the root-mean-square error (RMSE) of our model to be 8.02 (mean of prediction: 34.9).

The robustness of our model was tested by splitting the sampling sites into mountainous sites (elevation >100 m, 19 sites) and the remaining sites in the plains (20 sites). Then, we fitted the same model to the two data subsets (Table S3).

We also plotted the residuals of the model against terrestrial LULC effects and the model prediction (Fig. S12), and we did not find any obvious trend in these scatter plots, proving the normality assumption in model optimization. In addition, a normal Q-Q plot for the residuals was given to further support this assumption (Fig. S13). To estimate the uncertainty of parameters, we calculated profile likelihood-ratio confidence intervals (CI) of levels of 50% and 90% for each model parameter (see Supplementary Text; Table S8).

## Modeling terrestrial LULC effects on fish species distributions (species-level modeling)

To predict the habitat distribution of fish species, we generalized our model to a species-level model by modifying the fish species richness observation of Eq. 5 with a logit function. Specifically, we substitute $B$ with $ln(P/1-P)$, where $P$ is the probability of presence of a fish species in Eq. (6).

$$\ln\left(\frac{P}{1 - P}\right) = a + b \cdot \ln Q + C(r) \cdot V + \varepsilon. \quad (6)$$

Whereby, $\varepsilon$ is an $M$-vector of errors, in which each element is assumed independent and identically normally distributed in this case. Then, we applied a maximum likelihood estimation to find the effective spatial range $r$ and magnitude $V$ of terrestrial LULC effects. The associated LULC for each species was assigned by the LULC type with the highest terrestrial LULC effect ($V_k$) value.

## Terrestrial LULC effect map

We mapped the LULC effect on fish species richness for each terrestrial pixel ($E_{LULC_j}$) by tracing the flow path of pixel $j$ according to the flow direction map and summing up its LULC effect along the major river channels downstream (see Fig. S7 for a schematic diagram). The flow path of pixel $j$ comprises of two sections: the terrestrial path ($L_{ter,j}$, unit: km) and the river path ($L_j$, unit: km); and the integral of $V_{LULC_j}$ (derived from $V$ depending on the LULC type of pixel $j$, unit: num. species / km$^2$) along river path ($L_j$) starts from where water flow tracing of pixel $j$ entering the major river channels. To simplify computation, the effective distance $r$ (unit: km) was set as the upper boundary of the integral (i.e., $r = L_{ter,j} + L_j$).

Hence, the value (i.e., $E_{LULC_j}$) in the map can be directly perceived as the change of fish species richness per km in the river (unit: num. species/km) due to the terrestrial LULC effect from pixel $j$ (Eq. 7; see Fig. 3a).

$$E_{LULC_j} = V_{LULC_j} \cdot \int_{s=L_{ter,j}}^{L_{ter,j}+L_j} \frac{3}{r} e^{-\frac{3s}{r}} ds = V_{LULC_j} \cdot e^{-\frac{3L_{ter,j}}{r}} \cdot \left(1 - e^{-\frac{3L_j}{r}}\right). \quad (7)$$

Based on the previous bootstrapped samples, we predicted the terrestrial LULC effect map 2000 times and then calculated the interquartile range (IQR) as a metric of uncertainty (Fig. S14a, c).

## Neutral meta-community (NMC) model as a null model

We compared our results with simulations based on a null model of a quasi-neutral river meta-community model, which considers climate, fish habitat capacity, speciation, extinction, migration, and river network structure[31]. This model uses meta-community theories and fish dispersal in the riverine network to predict fish species richness patterns. In the NMC simulation, the product of average annual runoff production (AARP) and watershed area (WA) was replaced by river discharge ($Q$) acquired from the Hydro-SHEDS data, as they represent similar meaning and have high correlations. We ran 30,000 random simulations with different sets of parameters over a meaningful range and derived the set of parameters that mostly fitted the fish species richness pattern and had the highest adjusted $R^2$ (Table S4). Then, we compared its prediction error pattern with the error pattern of the FishDiv-LULC model (Fig. S6).

## Projecting a riverine fish species richness map

We applied our model to major river channel pixels in the catchment to generate a riverine fish species richness map (Fig. 3b). To do so, we produced a local catchment within a 19 km spatial range for each major river channel pixel, followed by applying our FishDiv-LULC model to predict fish species richness in the river. We also assessed the uncertainty by predicting fish species richness based on the estimated parameters from 2000 bootstrapped samples and then computed the IQR of prediction results for major river channel pixels (Fig. S14b, d).

## Fish traits in relation to terrestrial LULC

We collected ten major fish morphological traits from the FISHMORPH database[34]. They are maximum body length (MBl), body elongation (BEl), vertical eye position (VEp), relative eye size (REs), oral gape position (OGp), relative maxillary length (RMl), body lateral shape (BLs), pectoral fin vertical position (PFv), pectoral fin size (PFs), and caudal peduncle throttling (CPt), relating to fish metabolism, hydrodynamics, body size and shape, trophic levels and impacts, etc. Then, we linked fish traits and associated LULC type for each species according to the largest positive LULC effect value in the species-level modeling result. Species that could not establish species-level models were removed, so 93 out of 108 species were finally analyzed. Lastly, we mapped trait envelopes of LULC-associated fish species using a principal component analysis of ten morphological trait space.

## Fish species richness changes in the past and future

To predict fish species richness patterns in the past and future, we used the ESA CCI land cover map in 1992 (the first year of the product) and the GLOBIO4 land use maps in 2050 as past and modeled future LULC maps, respectively[23,37]. The GLOBIO4 2050 land use maps are predicted based on the present ESA CCI land cover map, showing good consistency with the global LULC product used in our modeling. For 2050, we used three LULC maps under shared socio-economic pathway 1 representative concentration pathway 2.6 (SSP1 RCP2.6), SSP3 RCP6.0, and SSP5 RCP8.5 scenarios.

Due to the lack of differentiation between rainfed cropland and irrigated cropland in the future maps, we merged these two LULC types (four LULC types in total) and refitted the model to predict fish diversity changes in the past and future. The new validation is shown in Table S9. We predicted riverine fish species richness maps of 1992 and 2050 (under three scenarios), and then calculated the percentage of changes in the periods of 1992–2016 (past) and 2016–2050 (future) (Figs. 4 & S9).

## Distribution changes of fish species of conservation concern (SPCC) in the past and future

We predicted the distribution map for each SPCC by firstly calculating a probability map in major river channels, and afterwards, determining presence/absence at each river channel pixel by a threshold with the highest true skill statistic value.

To assess the effects of LULC changes on SPCC, we also predicted the distribution maps for SPCCs in the past (1992) and forecast future fish distribution changes under three LULC change scenarios (2050). As a result, the distribution changes of SPCCs from 1992 to 2016 (past) and under three scenarios from 2016 to 2050 (future) are depicted in Figs. 4 & S9.

## River water properties from remote sensing data

We estimated river water properties of chlorophyll-a content (Chl-a), total suspended solids (TSS), and dissolved organic carbon (DOC) to show nutrient/resource availability in rivers. To improve accuracy, image collections of Sentinel-2 level 2A surface reflectance data were used to obtain a 20-m cloud-free image on the Google Earth Engine. Then, we extracted major river channel pixels using a water occurrence map from the 30-m global surface water data with a threshold of 0.75, which effectively filtered out most river shoreline pixels[53]. After that, non-river-channel and narrow-channel (<~60 m) pixels were carefully removed manually, and the surface reflectance image was resampled to match the resolution of water occurrence data. Next, the dominant LULC type for each river pixel was determined within a 4-km circle. We computed Chl-a, TSS, and DOC for river pixels following well-established methods[54–56], and plotted the water property values for dominant LULC types (Fig. 2). Detailed formulas of Chl-a, TSS, and DOC calculation can be found in the Supplementary Text.

## Reporting summary

Further information on research design is available in the Nature Portfolio Reporting Summary linked to this article.

## Data availability

Sequence data that support the findings of this study have been deposited in the European Nucleotide Archive under the primary accession numbers PRJEB34331 (Kelly dataset) and PRJEB34332 (MiFish dataset).

## Code availability

Code for computing the catchment and FishDiv-LULC model has been deposited at Github (https://github.com/hengzhang-zh/FishDiv-LULC-Model).

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

## Acknowledgements

We especially thank Yixin Hao in space physics for his invaluable insights and inspirations to our FishDiv-LULC modeling framework. We thank Elvira Mächler for her help with eDNA sample processing, Michael O'Brien, Luca Carraro, and Helen Kurkjian for their help with revising the manuscript. We thank Fernando Pelicice and two anonymous reviewers for their comments on the manuscript. F.A. is funded by the Swiss National Science Foundation Grants No. 31003A_173074 and 310030_197410, and F.A., R.F., and M.S. are further supported by the University of Zurich Research Priority Programme on Global Change and Biodiversity (URPP GCB).

## Author contributions

F.A. and H.Z. designed the research; H.Z. performed the research and developed the model; R.F. contributed to statistical methods; M.O. conducted the fieldwork; R.C.B., M.O., J.B., C.D.M., L.R.H., and B.H. did the bioinformatic analysis; H.Z. and F.A. wrote the original draft; H.Z., R.C.B., R.F., M.O., J.B., C.D.M., L.R.H., B.H., P.A.N., L.P., M.E.S., S.Z., and F.A. contributed to reviewing and editing the text.

## Competing interests

The authors declare no competing interests.
