## [Transparent Peer Review file · Communications Biology]

Terrestrial land cover shapes fish diversity in a major subtropical river catchment

Corresponding Author: Professor Florian Altermatt

This manuscript has been previously submitted at another journal. This document only contains information relating to versions considered at Communications Biology.

Version 0:

Reviewer comments:

Reviewer #1

(Remarks to the Author)

Paper: Terrestrial land cover shapes fish diversity in major subtropical rivers
By Heng Zhang et al.

The Zhang et al. manuscript presents a very interesting study using a new approach that recognizes the spatial structure and functioning of watersheds, the spatially structured land use, and how both interact to determine the proximity of land use effect on fish. I'm unaware of any other study that used such an approach or similar and I do think this is an important contribution to the field. For example, this could tell us about how far away we should keep land use from the river to reduce their impacts on fish. I have some comments and concerns below, both minor and major, which I think could be clarified or addressed by the authors to make the manuscript even more compelling and straightforward.

Title: "Terrestrial land cover shapes fish diversity in major subtropical rivers".

I would recommend choosing a title that both better highlights the main novelty of your study and also that clarifies that you are not assessing "major subtropical rivers across the world". Instead, you are using a case study of a single watershed in a single subtropical region for this.

80-83: I don't agree with such statements, there are several studies in the tropics (Brazil) that assess LULC at catchment scale quantitatively....

86-88: Yes! That is a key point of your study! I agree with this statement.

99-101: I think it is important to emphasize that this scenario is happening globally, and place your study in a global context, as a case study it is.

103: Novelty: Besides using (eDNA)-based fish diversity.... What else? Please state clearly in this paragraph what is the main advancement your study is providing to the field given previous studies relating fish diversity to land use in tropical/subtropical regions of the world? There are plenty of studies analyzing such relationships.

Results:

This is a very long result section, making it difficult to follow. Please, synthesize it. This may be due to the amount of discussion presented mixed with results. I think that mixing such an amount of discussion with results makes it confusing for the readers.

126-127. Did you include non-native species in your analysis?

132-134: Could you show a test for this claim?

135: 300m is a pretty coarse resolution for land use land cover in your scale of study. There are LCLU images for the world of 30m resolution nowadays. I think the 300m could be an important source of "error" or uncertainty in your results, mainly considering that the main goal of your study is to assess the effects of LCLU spatial structure and configuration on fish. This would also be a major problem to detect important riparian forest fragments, most of which are usually narrow (10-30 m wide).

For example, see: <https://glad.umd.edu/dataset/GLCLUC2020>

Potapov, P., Hansen, M. C., Pickens, A., Hernandez-Serna, A., Tyukavina, A., Turubanova, S., ... & Kommareddy, A. (2022). The global 2000-2020 land cover and land use change dataset derived from the Landsat archive: first results. *Frontiers in Remote Sensing*, 3, 856903.

136: provide the website for where the data was downloaded from.

146-150: I did not fully comprehend how the flow direction calculation was done for each site and how and what it was used for. I think this needs to be much better explained, both here and in the methods. It seems it is supposed to be some kind of weight used to combine with land use, but I'm not sure.

151: I did not understand "to 31—1,076 km upstream". Is it 31 corresponding to 1,076 km²? How is that so?

163-164: We need a better explanation about how the authors calculated this effect.

166-167: Move your sentence "Rainfed cropland had a significant relative positive effect on fish species richness, whereas forest and urban areas had relatively negative effects (Table 1)." to the below paragraph, otherwise it is not possible to follow the rationale for the next paragraph.

175-179: I don't know if this is usual to this journal, but there is a lot of discussion mixed with results. Not only here, but in the entire Results section. Apologize if this is something required by this journal, otherwise, I would recommend leaving all discussion to the discussion section only, and just state your findings here. For example, if you present some comments about results here, I was wondering why there is no discussion regarding other important findings presented above. So to avoid this, I would present almost all discussion in the discussion section only.

174: I don't see a direct connection between fig.2 and the statement presented in the above sentences, as you are talking about community size and species richness and your fig2 is traits space distribution. Otherwise, better explain how fig.2 results relate to your claims.

188-193: By the results presented in table S3, I did not come to the same conclusion as claimed in those sentences, as there is no support for an effect of Crop (R) in mountain sub sample sites ($P = 0.293$), only in plain sites ($P = 0.012$). Apologize if I did not interpret your result table correctly. Also, please, explain in Table S3 legend what are the parameters you are presenting there.

205-225: This is an interesting analysis, but I have the impression that it is not adding much to your principal objective. I mean, most of your conclusion, if not all of it, regarding the fish species richness and composition and how they relate to land use configuration (weighted by flow direction?) do not change with the inclusion of this trait envelope analysis by land use type. I would recommend removing this part from this manuscript for consistency purposes, and perhaps use this analysis in another manuscript addressing trait-based questions.

235: Fig. S7 is very important to understand your approach. I would move it to the main text, in the methods. I still do not fully comprehend how Fig. 3a is produced in terms of how this data is obtained and how the values are generated, and also how to interpret the figure. Perhaps because it is a new approach, authors should explain a little better along the text. Removing the trait-based approach would provide you more room to address this recommendation.

227: These projections are very interesting and should be addressed briefly elsewhere before in the text, perhaps in the introduction, abstract and in the title either. Notice, again, that you did not need your trait-based analysis to do these projections.

280: About eDNA samples. I wonder if it is already known how much of the upstream water (full of eDNA) influences your local samples. Is this already known? Because I would "guess" that fish species dna inhabiting further upstream reaches (let's suppose, perhaps something around 1km upstream) could be captured in the water you sampled in each site. Is this rationale correct? I apologize if not.

305-307: I agree! You may want to mention the native invasion phenomena. This may be interesting to explain your patterns. See: Scott, M. C., & Helfman, G. S. (2001). Native invasions, homogenization, and the mismeasure of integrity of fish assemblages. *Fisheries*, 26(11), 6-15.

Discussion:

I consistently agree with the discussion about the findings. However, I missed a paragraph addressing specifically the 19km upstream land use finding and how this value relates to previous findings from other studies investigating fish and land use (there are plenty), and what is the implication of this finding to policy and future research.

Methods:

393: Is it possible to estimate how many species were not included or not detected in a given site using these methods just because they are not in the DNA database? How much impact would this have in your results? Please, notice that many people that will read your paper will not be familiar with your methods for detecting ("collecting") species and will not go after other papers to learn about your method, so I think you need to be clear about this in your paper.

397-398: Would it be better if you used the average species number across the two methods, since both methods are incomplete and you don't know which one is best?

409: This 300m resolution satellite images might be an issue for detecting riparian forests, which are usually narrow (<50m wide). If so, then I would not be able to detect near stream land use effects on fish. Is this a concern? If so, how have you dealt with it?

421-427: This is a very interesting approach that makes much sense for me, congratulations.

436-480: Whereas it seems to be a robust analysis/way of calculating the parameters, I have to say that I don't quite fully comprehend it. So I wonder if the authors could improve the explanation here.

527: I think figure S7 should be placed in the main text.

Fig.3: Please, use different colors in the A and B maps, as the purple color for estimated richness in map B confused me with the purple color used to depict the land use-spatial effects in map A.

560-571: I don't think this trait-based analysis is adding much to your paper, it does not seem to be the main focus of your study and you already have too many analyses and approaches going on in our manuscript. Actually, I think the trait-based approach deserves to be better explored if expanded in another paper. For instance, investigate how the effects of spatially structured land use change when addressing different traits.

Reviewer #2

(Remarks to the Author)

Dear authors and Editor

Thank you for the opportunity to review this manuscript.

The study investigated the effects of land use on fish diversity in the Chao Phraya catchment, Thailand, employing sophisticated and robust approaches (e-DNA, spatial modeling). The topic is highly relevant considering the current losses of natural vegetation and its effects on adjacent river ecosystems and biodiversity – a global process.

The study is well designed, the manuscript is clear and well written, and results valuable for a wide audience, with significance for biodiversity conservation policies.

My review is positive. Yet, I have one main concern, in addition to some questions and comments. Please, check my report below.

I applaud this initiative.

Fernando Pelicice (UFT, Brazil)

MAIN CONCERN

I see two potential limitations with data collection (fish e-DNA) and the study design. I think they deserve some consideration. Please, consider the following:

- data sampling (e-DNA) has temporal restrictions. While the study covered the whole drainage, samples were collected in the dry season only, once in each site. They look like a snapshot sampling, and may not represent a broad and solid characterization of local fish assemblages. I encourage authors to explicitly address this potential limitation.

- fish diversity was based on e-DNA samples collected in the environment. These samples were used to characterize local species richness at each site. Yet, how do you know that the detected species is present in the respective site? Presumably, DNA fragments in lotic systems drift in the downstream direction, so there is a chance of recording a species absent in the site (but present somewhere upstream). It seems to be a major limitation of the study, as it is based on correlations between local richness and the surrounding environment. I encourage authors to justify how this limitation has affected results, and how it could be minimized.

TITLE

The title conveys the false idea that multiple subtropical rivers were studied. The study focused on a single catchment as a model. For coherence, the title needs revision.

RESULTS

Results are mixed with methods and discussion in some parts. A brief presentation of methods is understandable, but explanations and interpretations seem out of place. For example, check lines 168-179. Please, check if it is a problem.

Fish trait analysis was presented as Validation (In. 181 section). It does not look like a validation procedure. I think it should be presented as a main result, incorporated as a main objective.

The interpretation of fish trait patterns must consider that a few species may have strong effects on the functional volume. The description (In. 211-221) is simplistic and disregarded this aspect. For example, text says "Fish species associated with rainfed cropland had high body elongation (BEI), high ranges of relative maxillary length (RMI), and body lateral shape (BLs), indicating better hydro-dynamics". This is not the main pattern if we look at Fig 2, as most species clouded in the center of the volume. Perhaps the centroid is similar for all LULCs (it could be tested), although the incidental presence of a few different traits causes strong effects on volume area and shape. Please, consider revising the text and interpretations.

DISCUSSION

The curious increase in species richness in impacted areas was carefully interpreted, and the paper correctly acknowledged past investigations. Yet, I would avoid some common-sense expectations like "it did not suggest that cropland expansion would always benefit fish biodiversity protection" (In. 312). Obviously, increases in richness must not be associated per se with biodiversity conservation – context is required to interpret any relationship. Therefore, I suggest interpreting increase/decrease trends (or positive/negative effects) in terms of biodiversity "change". It is a more neutral, less prone to cause confusion or misinterpretation.

Another relevant point is fauna homogenization due to habitat simplification. The loss of habitats and spatial heterogeneity is severe and may induce local gains but regional losses. The topic was barely touched, and I think it could be better discussed in the context of the present investigation (expansion of croplands).

An important open question about stream integrity is the relative importance of the riparian buffer versus the catchment. The riparian buffer is well recognized, but distant effects from land use changes has not been evoked (or demonstrated) as a main driver of local changes. The present study is focused on this issue, but it was not emphasized. This topic is important from a management point of view, because legislation usually address each scale differently. In Brazil, for example, the protection of the riparian buffer is mandatory, but legislation is more permissible about the removal of vegetation in the catchment beyond the river channel. This scenario is relevant.

A main stressor is river fragmentation. Although this stressor was commented in the Introduction section, the Discussion ignored it. Consider addressing its potential use in spatial modeling coupled with LULC analysis.

Reviewer #3

(Remarks to the Author)

This manuscript analyses the relationship between Land Use-Land Cover (LULC) and freshwater fish species richness and composition across a large river catchment from Thailand using eDNA samples to detect species presence and estimate richness values over the catchment. Although the manuscript is very well written and follows a clear and logical line and framework, I found quite difficult to understand in depth what the analyses are offering and what is the main message and value of the outputs found. The authors use a very complex methodology, which I acknowledge that I am not familiar with, but seems anyway too much in face of the general target, which is to relate two "simple" parameters, richness and LULC. Overall, their seems to be great creativity and novelty in the methodology, however I see no great novelty in the aims and results, that would justify publishing in *Communications Biology*, although that is an editorial decision, not a reviewer one. Maybe the manuscript would better fit in for instance *Ecological Indicators*? There is clearly an enormous analytical work in this manuscript, and sound results, which deserves being published for sure. I would suggest the authors to either simplify or explain why such complexity is needed to analyze their data and also make a more clear point on what are the main findings and their importance.

These general concerns being said, I still have one methodological comment related to the number of species detected.

With eDNA sampling sites covering most of the catchment, authors detected around 100 species, while the freshwater fish richness in Thailand is estimated to be between 1000 and 1300, with circa 900 already described species, from which around a half may be present in the Chao Phraya basin. I would at least expect that the manuscript discuss this and explain why it's not an issue, or how it could affect the results.

Another point to address or clarify is about the resulting patterns, namely a negative relationship of richness with forest and a positive one with rainfed cropland. When looking at the map, forests are placed in upstream areas and cropland downstream. Even if river discharge is accounted for in the analyses, could this result be related to the well-known increase in fish species richness along the longitudinal gradient of rivers?

Version 1:

Reviewer comments:

Reviewer #1

(Remarks to the Author)

Most of my concerns were addressed or explained by the authors in this reviewed version.

The main limitation of the study of using 300-m resolution LULC images are difficult or even impossible to be resolved with the current data available worldwide, as claimed by the authors. I do think, however, the overall patterns of the results are still valid and consistent, although the exact effect size and distance-weighted LCLU values could change if a more refined resolution were available. On the other hand, I think authors could discuss this limitation briefly somewhere in the Discussion or in the Methods.

I have some addition comments, however:

Discussion

L258: the amount of forest and/or cropland and urban land use are usually negatively correlated. So I recommend being careful in saying that forest has a negative effect of richness. This is most likely a positive effect of cropland related to the factors already addressed by the authors, rather than a negative effect of forest. Please, review/rewrite such a statement here and elsewhere in the results and discussion.

301-322: The authors decided to maintain the trait-based approach in the manuscript. However, now they are presenting the results for this part in the Discussion. I do think since they decided to keep this approach they should present these results in the Results section and better use this paragraph to explain the meaning of such results.

323-342: I notice the authors included a discussion about homogenization as recommended by one reviewer. I recommend, however, to be careful when talking about homogenization, because some recent papers have been showing that human modifications can also cause differentiation of communities, which can be as bad as homogenization. The spatial and temporal scale addressed in the study and in the discussion will have an important implication on whether you will see homogenization or differentiation of communities. So please make it explicit what is that spatial (within river? between rivers? in the whole catchment?) and temporal scale (between years? or along the time?) you are discussing in this paragraph and also what is the diversity metric (taxonomic vs. functional vs. genetic? Species level? Family level?) you are talking about.

Reviewer #2

(Remarks to the Author)

Dear Authors

Thank you for revising the manuscript and addressing reviewers' comments. The original submission was already very well-done, and the revised version is even more robust and clear.

I have no further comment to provide. Congratulations for this project, it is a great contribution to the field.

I recommend publication.

Regards

Fernando Pelicice

Response to reviewers

Reviewer #1 (Remarks to the Author):

Paper: Terrestrial land cover shapes fish diversity in major subtropical rivers

By Heng Zhang et al.

The Zhang et al. manuscript presents a very interesting study using a new approach that recognizes the spatial structure and functioning of watersheds, the spatially structured land use, and how both interact to determine the proximity of land use effect on fish. I'm unaware of any other study that used such an approach or similar and I do think this is an important contribution to the field. For example, this could tell us about how far away we should keep land use from the river to reduce their impacts on fish. I have some comments and concerns below, both minor and major, which I think could be clarified or addressed by the authors to make the manuscript even more compelling and straightforward.

RESPONSE:

Thank you very much for recognizing and highlighting the potential of our work! We appreciate and value your comments, and agree that the FishDiv-LULC modeling framework can provide essential information for freshwater biodiversity conservation and land use management and has a broad range of applications. Indeed, as noted by you, our study is also to our knowledge the first study that systematically links terrestrial land use effects and river fish distributions in a predictive manner by explicit spatial range and magnitudes of the effects.

Title: "Terrestrial land cover shapes fish diversity in major subtropical rivers".

I would recommend choosing a title that both better highlights the main novelty of your study and also that clarifies that you are not assessing "major subtropical rivers across the world". Instead, you are using a case study of a single watershed in a single subtropical region for this.

RESPONSE:

Thanks for pointing this out. We used "in major subtropical rivers" because the Chao Phraya catchment comprises four tributaries (Ping River, Wang River, Yom River, and Nan River) and one main stem (mid- to lower reaches of Chao Phraya River), therefore five rivers in total. However, we recognized that this title might be somehow misleading. We have changed it to "in a major subtropical river catchment".

80-83: I don't agree with such statements, there are several studies in the tropics (Brazil) that assess LULC at catchment scale quantitatively....

RESPONSE:

We have changed the sentence. "Despite many efforts, it remains difficult to estimate the spatial extent and magnitude of terrestrial LULC effects on riverine fish communities, and the estimates are subject to

considerable uncertainty”. We have also changed the last sentences in the indicated lines. “yet they are individually assessed at local scales with high uncertainties and cannot be used to accurately predict fish diversity”. We have also added more references about Fish-LULC studies especially in the Brazilian Amazon.

86-88: Yes! That is a key point of your study! I agree with this statement.

RESPONSE:

Thanks for your positive comment! This is indeed the missing but crucial part in all the Fish-LULC analyses.

99-101: I think it is important to emphasize that this scenario is happening globally, and place your study in a global context, as a case study it is.

RESPONSE:

We do agree with you. There are many papers talking about the LULC change from the past to the future at the global context. In fact, we are currently conducting a global fish diversity-LULC analysis by applying this FishDiv-LULC model to other biomes worldwide and we found this model also worked well in many parts of the world. So, there is no doubt about the scalability of this model.

While we acknowledge the global scope, and adjusting accordingly, we preferred to keep the Introduction section centered around the subtropical river basins, to not imply a broader focus than we have, and because our study focus on them. However, we have now specifically pointed out the potential of scalability in the Discussion section.

103: Novelty: Besides using (eDNA)-based fish diversity.... What else? Please state clearly in this paragraph what is the main advancement your study is providing to the field given previous studies relating fish diversity to land use in tropical/subtropical regions of the world? There are plenty of studies analyzing such relationships.

RESPONSE:

Thanks for bringing this up! We have changed accordingly. Now we highlight the novelty much better.

Results:

This is a very long result section, making it difficult to follow. Please, synthesize it. This may be due to the amount of discussion presented mixed with results. I think that mixing such an amount of discussion with results makes it confusing for the readers.

RESPONSE:

Indeed, we had included some “discussion” into the Result section to fit the journal’s writing style. As both you and the second reviewer raised this point, we now moved the “discussion-like” results from the Result section to the Discussion, giving it a better flow and structure.

126-127. Did you include non-native species in your analysis?

RESPONSE:

The list of species names is in Table S1. According to the suggestion from the co-author who conducted the eDNA sampling, we removed two non-native species from the original nine IUCN species. The remaining IUCN species have their distribution not constrained within the Chao Phraya River but could be at a broader scale such as Southeast Asia or spatially adjacent rivers (e.g., Mekong River).

132-134: Could you show a test for this claim?

RESPONSE:

Yes, we are happy to do so. The eDNA fish species richness sampling result is provided in Figure 1. We now added an extra figure in the Supplementary Materials as follows:

Figure R1/S1 Percentages of area of five land use and land cover (LULC) types (irrigated cropland, rainfed cropland, forest, shrub- and grassland, urban area) in the buffer zone within a 6 km flow distance upstream of the mountainous sampling sites. We classified 19 sites into “sites near cropland” and “sites away from cropland” according to their percentages (> 60%) and distributions of croplands. The mean fish species richness of “sites near cropland” and “sites away from cropland” are 34.9 and 20.9, respectively, showing cropland

To make it more accurate, we classified all the mountainous sites into “sites near cropland” and “sites away from cropland” according to cropland locations (whether spatially close to river channels) and percentages in area within a 6 km flow distance catchment upstream. Specifically, we identified sites #1,

2, 4, 10, 12, 14, 16, 17, 18, 20, 25, and 26 as “sites near cropland”. Then, we computed the mean of observed fish species richness for the two groups and found the average fish species richness was remarkably higher at “sites near cropland” (mean = 34.9) than “sites away from cropland” (mean = 20.9). We have added essential information in the main text.

135: 300m is a pretty coarse resolution for land use land cover in your scale of study. There are LCLU images for the world of 30m resolution nowadays. I think the 300m could be an important source of “error” or uncertainty in your results, mainly considering that the main goal of your study is to assess the effects of LCLU spatial structure and configuration on fish. This would also be a major problem to detect important riparian forest fragments, most of which are usually narrow (10-30 m wide).

For example, see: <https://glad.umd.edu/dataset/GLCLUC2020>

Potapov, P., Hansen, M. C., Pickens, A., Hernandez-Serna, A., Tyukavina, A., Turubanova, S., ... & Kommareddy, A. (2022). The global 2000-2020 land cover and land use change dataset derived from the Landsat archive: first results. *Frontiers in Remote Sensing*, 3, 856903.

RESPONSE:

Thanks for bringing this up! We had indeed thought about using LULC map of higher resolution (e.g., 100 m to match HydroSHEDS flow direction map, or the 30 m database suggested by you). However, we finally chose the 300 m ESA CCI LC map because of the following reasons.

First, although the 300 m resolution LULC map might have higher uncertainties than a 30 m resolution map in terms of spatial modeling, we had actually found that the largest uncertainty came from catchment calculation. In our very preliminary analysis, we had generated flow direction map only based on the Shuttle Radar Topography Mission (SRTM) 30 m digital elevation model (DEM) data and then calculated the catchment ourselves (the river network and the shape of catchment are shown in Figure R2).

Figure R2 River network generated from SRTM DEM data using ArcMap Version 10.8. There were notable discrepancies in the structure and direction in the river networks especially in the lower reaches, which led to large uncertainties in modeling. An example is the river channel between site #21 and 22. According to the “ground truth” of Sentinel-2 images and Google Map, the river flows to the south to site #22 rather than southeast as calculated using SRTM DEM.

The adjusted R^2 for the model fitted with catchments derived from SRTM DEM was 0.369 (four-class system) showing much poorer result than the model derived from HydroSHEDS (0.582) which included a “ground truth” calibration process in the data production. Nevertheless, even a calibrated database like HydroSHEDS has non-negligible uncertainties/errors in the structure of river network with a resolution of

90 m. For example, the meandering river channels in the mid- to lower reaches are not absolutely precise with a spatial uncertainty up to several pixels. Given these reasons, we would use highly accurate flow direction map (also with finer resolution) rather than a very-high-resolution LULC map to improve the performance as the first step. Plus, to our knowledge, these hydro-maps are still under development.

Second, we chose the 300 m-resolution ESA CCI LC map because the future LULC projection maps (GLOBIO4) were based on ESA CCI LC data, which to our knowledge, was also one of the finest resolved and robust predictions. In a past-present-future comparison, such a consistence in spatial resolution needed to be kept. We would love to use 30 m resolution LULC map given there is a robust future LULC projection at the same spatial resolution. However, today this is still not possible.

Third, our number of sampling sites in the Chao Phraya River catchment limited modeling with high complexity. By having a 30 m resolution we would imply a precision that is not matched at the eDNA sampling scale (e.g., some of the river sections are wider than 30 m, and we collected the eDNA at both shore lines, thus, this would imply a precision that is not given). The 300 m resolution is more conservatively capturing and matching the precision of the eDNA sampling. Overall, we had to choose an optimal way to balance the reality of the world and the complexity of our spatial model. In this sense, adding much accuracy from the LULC perspective will only have a marginal effect. To reduce the uncertainty, having more sampling sites would have higher priority than using a very-high-resolution LULC map. These will be our future steps.

Lastly, another consideration is the limitation of computational resources. Essentially, we had aimed for a computationally friendly modeling solution that could be conducted with a local PC. As is shown in the “Data Availability” section, we programmed the catchment computation and the FishDiv-LULC model with CUDA C for GPU computing. In this study, the prediction of river fish species richness was calculated at the resolution of HydroSHEDS flow direction map (90 m) which was already time-demanding (almost 6 hours using NVIDIA RTX A5000 24 GB GPU). So, the computation time would be more than 10 times if the spatial resolution is downscaled to 30 m. Moreover, such computation of higher spatial resolution would require a much higher use of memory. We think that especially for applying our model to lower income regions, having computationally economic approaches is appropriate as it reduces barriers of local reproduction.

Given the above reasons, we would like to use the 300 m LULC map in this research. However, we will revisit the model using a 30 m LULC map when there is a good improvement in both the accuracy of hydro-maps and the efficiency of computational platforms.

136: provide the website for where the data was downloaded from.

RESPONSE:

We had provided the link in the Materials and Methods section with a subtitle “Land use and land cover data”. The website is <https://www.esa-landcover-cci.org/>.

146-150: I did not fully comprehend how the flow direction calculation was done for each site and how and what it was used for. I think this needs to be much better explained, both here and in the methods. It seems it is supposed to be some kind of weight used to combine with land use, but I'm not sure.

RESPONSE:

Basically, the water flow direction map describes the direction of water flow from the current pixel to the neighboring pixels, which is determined by the steepest gradient of the elevational difference. So, this map can be used to simulate how the water flows in the river catchment. With a flow direction map, for each sampling site, one can track all the pixels in the map and determine if one pixel ultimately flows through the sampling site (also can calculate the cumulative flow distance). Then, all the pixels that flow through the sampling site are connected as a catchment map for that sampling site. We have added a few sentences in the main text to better illustrate the catchment computation concept.

Once we have the catchment maps with flow distances, we use them in the distance decay framework (the parameter r in km) in the FishDiv-LULC model to better capture the spatial influence from upstream terrestrial LULC.

151: I did not understand "to 31—1,076 km upstream". Is it 31 corresponding to 1,076 km²? How is that so?

RESPONSE:

Here, "31—1,076 km upstream" means the range of maximal flow distance within all the site catchments. We found this sentence could be misleading, so have changed accordingly. Thanks for pointing this out.

163-164: We need a better explanation about how the authors calculated this effect.

RESPONSE:

We have already explained the whole estimation process in the Materials and Methods in detail. The LULC effect (E_{LULCj} in num. species/km²) is estimated through the FishDiv-LULC model. In brief, we assume that the observed fish species richness is associated with the integral of LULC effects (the sum of effects of croplands, forests, etc.) within the site catchment times a distance decay framework (the exponential decay with a distance parameter r in km). Then, all the parameters including the LULC effects and effective distance and their uncertainties can be estimated by the maximum likelihood estimation (MLE).

Because the whole process is a little bit complex for a general reader who usually cares about the final results, we suggest interested readers to refer to the Materials and Methods section.

166-167: Move your sentence "Rainfed cropland had a significant relative positive effect on fish species richness, whereas forest and urban areas had relatively negative effects (Table 1)." to the below paragraph, otherwise it is not possible to follow the rationale for the next paragraph.

RESPONSE:

This is a very good piece of advice. We have changed accordingly.

175-179: I don't know if this is usual to this journal, but there is a lot of discussion mixed with results. Not only here, but in the entire Results section. Apologize if this is something required by this journal, otherwise, I would recommend leaving all discussion to the discussion section only, and just state your findings here. For example, if you present some comments about results here, I was wondering why there is no discussion regarding other important findings presented above. So to avoid this, I would present almost all discussion in the discussion section only.

RESPONSE:

We have responded above. The main reason was that we adopted a Nature formatting style rather than the traditional "introduction-method-result-discussion" logic. As this was not the most straightforward way of presenting, we have moved the discussion-type text to the Discussion.

174: I don't see a direct connection between fig.2 and the statement presented in the above sentences, as you are talking about community size and species richness and your fig2 is traits space distribution. Otherwise, better explain how fig.2 results relate to your claims.

RESPONSE:

Actually, here we referred to the bar plots in the right panels of Figure 2, which indicated the relative concentrations of Chl-a, TSS, and DOC. We have added "bar plots" in that sentence for better clarity.

188-193: By the results presented in table S3, I did not come to the same conclusion as claimed in those sentences, as there is no support for an effect of Crop (R) in mountain sub sample sites ($P = 0.293$), only in plain sites ($P = 0.012$). Apologize if I did not interpret your result table correctly. Also, please, explain in Table S3 legend what are the parameters you are presenting there.

RESPONSE:

Indeed, p-values were not small enough to make a very solid conclusion because of the relatively small number of sampling sites in each category. And this was why we never used "significant" in the sentence. Nevertheless, this comparison is still an important aspect in this study as it testified a nearly congruent (i.e., same positive/negative) LULC effects in both mountainous and plain areas. For better understanding, we have added "approximate" to indicate this was a rough estimation. We have also changed the following sentence accordingly.

205-225: This is an interesting analysis, but I have the impression that it is not adding much to your principal objective. I mean, most of your conclusion, if not all of it, regarding the fish species richness and composition and how they relate to land use configuration (weighted by flow direction?) do not change with the inclusion of this trait envelope analysis by land use type. I would recommend removing this part from this manuscript for consistency purposes, and perhaps use this analysis in another manuscript addressing trait-based questions.

RESPONSE:

Thanks for pointing this out. We respectfully think this is an important part of the research because, together with river water properties (Chl-a, TSS, DOC), it provided a mechanistic understanding about how fish species richness/distribution was shaped by nearby terrestrial LULC. It is quite natural for the readers/audience to ask why when they see this interesting spatial association and the modeling results. Simply to answer this question, we explored these fish traits that had specific functional meaning and found that fish traits played an important role in linking LULC and fish distributions. Moreover, this trait-based analysis was highlighted in the previous review and the novelty was highly valued. Consequently, such an analysis added more dimensions to the whole study. If this section were removed, we feel the entire study would be only statistical and correlational and less convincing. However, we have moved this section (together with river water properties) to the Discussion for a better logic line.

235: Fig. S7 is very important to understand your approach. I would move it to the main text, in the methods. I still do not fully comprehend how Fig. 3a is produced in terms of how this data is obtained and how the values are generated, and also how to interpret the figure. Perhaps because it is a new approach, authors should explain a little better along the text. Removing the trait-based approach would provide you more room to address this recommendation.

RESPONSE:

We agree that original Fig. S7 is a very important figure to illustrate how terrestrial LULC effect map was computed. Nevertheless, compared with the result figures in the main text, this figure could be somehow less important. In addition, the Nature writing style does not allow many figures and tables in the main text. Therefore, as a balance, we would keep the original Fig. S7 in the Supplementary Materials.

We have added more details about how to compute a terrestrial LULC effect map in that paragraph.

227: These projections are very interesting and should be addressed briefly elsewhere before in the text, perhaps in the introduction, abstract and in the title either. Notice, again, that you did not need your trait-based analysis to do these projections.

RESPONSE:

Thanks for highlighting our novelty! We do agree that this is a fundamental advance in biodiversity sciences in terms of biodiversity predictions. We have added this piece of information in the last paragraph of the Introduction section.

280: About eDNA samples. I wonder if it is already known how much of the upstream water (full of eDNA) influences your local samples. Is this already known? Because I would “guess” that fish species dna inhabiting further upstream reaches (let’s suppose, perhaps something around 1km upstream) could be captured in the water you sampled in each site. Is this rationale correct? I apologize if not.

RESPONSE:

Thanks for highlighting our novelty! Basically, yes. eDNA enters the river and flows with water before

decomposition, thus carrying signals of presence downstream. When taking water samples, the eDNA inside reflects the cumulative biodiversity information from the sampling site to a distance upstream. According to our previous study, which was carried out in a middle-size mountainous catchment in Switzerland, we found the maximum detectable distance was around 10 km upstream (Deiner & Altermatt, 2014), which is congruent and at the upper end of signal detection. In the subtropical catchment with higher temperature and more precipitation, the degradation of eDNA usually happens faster, leading to a smaller transport distance (usually a few km at maximum). Studies in the past a few years has shown good results of using eDNA for fish sampling (Pont et al., 2018). For example, a study in Japan has shown an 86.4 % congruence of eDNA and conventional sampling and also the distribution pattern for fish species (Nakagawa et al., 2018). Another study has revealed the detection upper extent of fish eDNA sampling in forested streams to be 50-250 m, showing a very good spatial consistency with traditional electrofishing method (Penaluna et al., 2021). So, one can reasonably assume that the eDNA samples reflects the local biodiversity information in the river channel.

305-307: I agree! You may want to mention the native invasion phenomena. This may be interesting to explain your patterns. See: Scott, M. C., & Helfman, G. S. (2001). Native invasions, homogenization, and the mismeasure of integrity of fish assemblages. *Fisheries*, 26(11), 6-15.

RESPONSE:

Thank you for providing this really interesting paper! This is indeed a potential mechanism that could explain the diversity gradient in the Chao Phraya River catchment. We have added this reference in the main text.

Discussion:

I consistently agree with the discussion about the findings. However, I missed a paragraph addressing specifically the 19km upstream land use finding and how this value relates to previous findings from other studies investigating fish and land use (there are plenty), and what is the implication of this finding to policy and future research.

RESPONSE:

Thank you for your comment. Reviewer #2 also mentioned this aspect of application of our research. Indeed, we totally agree that this is an important aspect in our study. We have expanded the last paragraph in the Discussion to demonstrate the meaning of the detected spatial distance and also the implication to the conservation practices. Particularly, we have explicitly added sentences about how our approach could help conservation planning in Amazonian countries where riparian buffer of tens of meters is adopted (still far from sufficient). Because it is a general topic of conservation planning, we did not mention the specific number (19 km) in that paragraph but focused on the logic behind.

Methods:

393: Is it possible to estimate how many species were not included or not detected in a given site using these methods just because they are not in the DNA database? How much impact would this have in

your results? Please, notice that many people that will read your paper will not be familiar with your methods for detecting (“collecting”) species and will not go after other papers to learn about your method, so I think you need to be clear about this in your paper.

RESPONSE:

This is a crucial point because such a mismatch is always a source of uncertainty for both traditional and eDNA sampling. Actually, our research lab had conducted a meta-analysis to compare the sampling results between eDNA metabarcoding and traditional methods (Keck et al., 2022). We found that fish sampling showed a good congruence in both approaches, which was much better than all the other compared taxa (plankton, microphytobenthos and macroinvertebrates). In addition, we took six samples (two samples in the middle of channel, two on the right and left side, respectively) with a relatively large volume (600 mL). In the eDNA sequencing, we adopted two primers, i.e., Kelly and MiFish, built a customized fish eDNA reference database, and merged the results by taking the higher read counts for each fish taxon. In summary, we took all the possible strategies to improve the detection ability using eDNA metabarcoding, which was significantly better than most of the eDNA studies used in the aforementioned meta-analysis. So, we were confident that the fish diversity pattern in this study was robust and had a good comparability.

397-398: Would it be better if you used the average species number across the two methods, since both methods are incomplete and you don't know which one is best?

RESPONSE:

We have explained in the above paragraph. Basically, Kelly and MiFish primers have different sensitivity to fish species. In this study, we took the higher read counts simply to improve the overall detection ability, which means that one fish species was present as long as it was detected by at least one primer. Given this reason, we would prefer to keep our original approach. This is also congruent to the technical paper on the eDNA sampling (Blackman et al., 2021).

409: This 300m resolution satellite images might be an issue for detecting riparian forests, which are usually narrow (<50m wide). If so, then I would not be able to detect near stream land use effects on fish. Is this a concern? If so, how have you dealt with it?

RESPONSE:

We have also explained why we used the 300 m resolution LULC map above. The main reason is that high uncertainties came from the hydro-maps (regarding catchment calculations) rather than LULC. We used the 90 m HydroSHEDS flow direction map to improve the modeling performance but it still brought uncertainties into our analysis. So, we would use a very-high-resolution LULC map once the high-resolution and reliable hydro-maps are ready. In this study, we can only focus on the overall effect and big patterns and would unfortunately ignore this level of details given the data limitations.

421-427: This is a very interesting approach that makes much sense for me, congratulations.

RESPONSE:

Thanks for your compliment! We used the conventional approach, i.e. D8 algorithm, to calculate the catchment and programmed it with CUDA C for GPU computing to cut the computational time to a few seconds for each site (otherwise would take hours). Please refer to our code in the Data Availability section.

436-480: Whereas it seems to be a robust analysis/way of calculating the parameters, I have to say that I don't quite fully comprehend it. So I wonder if the authors could improve the explanation here.

RESPONSE:

It is indeed technical because of the spatial statistical methods used here. Although we had tried to keep the method as simple as possible, it was unavoidable that a general reader needed to spend some time to fully understand these formulas (they were double-checked by our co-author in mathematics). We have expanded the paragraphs with details and more explanations to help the reader to better understand the logic of modeling.

527: I think figure S7 should be placed in the main text.

RESPONSE:

We have explained in the above paragraph. In brief, this schematic diagram figure could be somehow less important compared with the result figures in the main text, though we agree that it would be very helpful for general readers. Thus, we prefer to keep it in the Supplementary Materials.

Fig.3: Please, use different colors in the A and B maps, as the purple color for estimated richness in map B confused me with the purple color used to depict the land use-spatial effects in map A.

RESPONSE:

We had chosen this bluish purple because of the balance of color across the two panels (we had tried all the default color bars in the ArcMap software). Now we have changed the color bar to Blue-Yellow-Orange instead.

560-571: I don't think this trait-based analysis is adding much to your paper, it does not seem to be the main focus of your study and you already have too many analyses and approaches going on in our manuscript. Actually, I think the trait-based approach deserves to be better explored if expanded in another paper. For instance, investigate how the effects of spatially structured land use change when addressing different traits.

RESPONSE:

We have explained this point in detail in the above paragraphs. Briefly, we got this as a common question from the audience when we presented the modeling results, and were regularly inquired about

if this can be linked to traits of the fish communities. Thus, we conducted this analysis and provided important mechanistic understanding about how LULC could shape fish distributions. As such, we feel that it added essential dimensions to the whole study, yet have now moved it largely to the Discussion section.

Thank you again for your valuable comments which helped us to improve the clarity and readability of this manuscript!

Reviewer #2 (Remarks to the Author):

Dear authors and Editor

Thank you for the opportunity to review this manuscript.

The study investigated the effects of land use on fish diversity in the Chao Phraya catchment, Thailand, employing sophisticated and robust approaches (e-DNA, spatial modeling). The topic is highly relevant considering the current losses of natural vegetation and its effects on adjacent river ecosystems and biodiversity – a global process.

The study is well designed, the manuscript is clear and well written, and results valuable for a wide audience, with significance for biodiversity conservation policies.

My review is positive. Yet, I have one main concern, in addition to some questions and comments. Please, check my report below.

I applaud this initiative.

Fernando Pelicice (UFT, Brazil)

RESPONSE:

Thank you very much for your positive and constructive feedback as well as detailed comments! We are happy to see that you recognized the novelty of this project and the newly developed modeling framework. With your detailed comments and suggestions, we have carefully revised our manuscript. We also rewrote some paragraphs in the Result and Discussion sections to make it easier to understand for general readers.

We believe that our substantial improvements have enabled better clarity for general readers. Once again, we thank you for your valuable comments on our manuscript!

MAIN CONCERN

I see two potential limitations with data collection (fish e-DNA) and the study design. I think they deserve some consideration. Please, consider the following:

- data sampling (e-DNA) has temporal restrictions. While the study covered the whole drainage, samples were collected in the dry season only, once in each site. They look like a snapshot sampling, and may not represent a broad and solid characterization of local fish assemblages. I encourage authors to explicitly address this potential limitation.

RESPONSE:

Thank you for this comment. Indeed, we have one single timepoint for the extensive eDNA biodiversity survey. While more timepoints in all such studies are desirable, we start acknowledging that having biodiversity data from such subtropical catchments is not a given; thus, we want to explicitly demonstrate that even one sampling campaign can give sufficient data for a LULC effect comparison, yet obviously agree that more timepoints are always desirable. Nevertheless, we ensure that the timepoint chosen was adequate and representative. We took all the water eDNA samples during the dry season (November, 2016) firstly because in wet seasons turbidity appears to reduce detection sensitivity and increase false negative detection of eDNA sampling (Egeter et al., 2018; Williams et al., 2017). Second, the influence of floods to fish species richness has been reported to be the same level of hydro-conditions and water quality, especially in the Chao Phraya River catchment (Tanaka et al., 2015). Therefore, fish species richness patterns captured during the rainy season would unavoidably include these flooding effects, adding more uncertainties in LULC effect assessments.

We do agree that there some fish species' distribution may not completely captures by sampling only one season, because some of the detected fish species may migrate through river channels across seasons. Nevertheless, the role of such migratory effects in determining overall fish species richness patterns is less obvious than the resource availability and water quality effects typically associated with terrestrial land use types, especially in a river system of such a large scale (about 1600 km from the upper mountains to the mouth of the sea) and with such a high level of anthropogenic interference. Plus, due to the limitation of costs and efforts in our eDNA sampling, we could not capture this seasonal variability in the field campaign.

We have added sentences describing the reasoning of one seasonal sampling in the Discussion section. We would be happy to explore the seasonal change of fish-LULC associations in the future projects.

- fish diversity was based on e-DNA samples collected in the environment. These samples were used to characterize local species richness at each site. Yet, how do you know that the detected species is present in the respective site? Presumably, DNA fragments in lotic systems drift in the downstream direction, so there is a chance of recording a species absent in the site (but present somewhere upstream). It seems to be a major limitation of the study, as it is based on correlations between local richness and the surrounding environment. I encourage authors to justify how this limitation has affected results, and how it could be minimized.

RESPONSE:

Thanks for your comment, this is a very good point, and actually indeed there is some transportation of eDNA, which is generally well studied. We have explained this point in the response to reviewer #1 who mentioned the similar concern. In brief, eDNA transports along with river water before degradation, carrying signals of fish presence downstream. Therefore, eDNA acquired from flowing water reflects the local and cumulative biodiversity information from the sampling site to a certain distance upstream (maximum detection distance). According to previous studies, the degradation of eDNA depends on the temperature, water quality, bioactivity, and abundance of fish species in the water. Actually, our lab carried out an assessment of maximum detectable distance in a middle-size mountainous catchment in

Switzerland and we found the maximum detectable distance was maximally up to 10 km upstream (Deiner & Altermatt, 2014), yet most signals are transported over shorter distances (Brantschen & Altermatt, 2023; Carraro & Altermatt, 2022). Studies in the past a few years has shown good results of using eDNA sampling for fish and other taxa (Pont et al., 2018; Yang et al., 2021). For example, a study in Japan has shown an 86.4 % congruence of eDNA and conventional sampling and also the distribution pattern for fish species (Nakagawa et al., 2018). Another study has revealed the detection upper extent of fish eDNA sampling in forested streams to be 50-250 m, showing a very good spatial consistency with traditional electrofishing method (Penaluna et al., 2021). In the subtropical catchment with higher temperature and more precipitation, the degradation of eDNA usually happens faster, leading to a smaller transport distance. Plus, the average flow distance between our sampling sites was around 50 km, which was far beyond the transport distance in terms of biodiversity pattern. So, one can reasonably assume that the eDNA samples reflects the local biodiversity information in the river channel.

In addition, to reduce the uncertainty in species detection, we took six samples (two samples in the middle of channel, two on the right and left side, respectively) with a relatively large volume (600 mL for each replicate). In the eDNA sequencing, we adopted two primers, i.e., Kelly and MiFish, built a customized fish eDNA reference database, and merged the results by taking the higher read counts for each fish taxon after removing very rare OTUs from each dataset (a very standard and robust way of eDNA processing).

Consequently, we had optimized the detection ability as well as the robustness of eDNA metabarcoding in biodiversity monitoring, which was significantly better than many of the eDNA fish studies. We have added some sentences in the Discussion section to openly address the aspect of transportation.

TITLE

The title conveys the false idea that multiple subtropical rivers were studied. The study focused on a single catchment as a model. For coherence, the title needs revision.

RESPONSE:

We have changed the title accordingly.

RESULTS

Results are mixed with methods and discussion in some parts. A brief presentation of methods is understandable, but explanations and interpretations seem out of place. For example, check lines 168-179. Please, check if it is a problem.

RESPONSE:

Thanks for your comment. It was also mentioned by the reviewer #1 who suggested that these kinds of analyses should move to the Discussion section. Therefore, we have restructured the Result and Discussion sections and moved these analyses to the Discussion. We hope the new arrangement/structure can help a general reader better understand our story.

Fish trait analysis was presented as Validation (ln. 181 section). It does not look like a validation procedure. I think it should be presented as a main result, incorporated as a main objective.

RESPONSE:

We have moved this analysis to the Discussion section.

The interpretation of fish trait patterns must consider that a few species may have strong effects on the functional volume. The description (ln. 211-221) is simplistic and disregarded this aspect. For example, text says “Fish species associated with rainfed cropland had high body elongation (BEI), high ranges of relative maxillary length (RMI), and body lateral shape (BLs), indicating better hydro-dynamics”. This is not the main pattern if we look at Fig 2, as most species clouded in the center of the volume. Perhaps the centroid is similar for all LULCs (it could be tested), although the incidental presence of a few different traits causes strong effects on volume area and shape. Please, consider revising the text and interpretations.

RESPONSE:

We had presented a quantitative table (Table S5) demonstrating the mean, range, and area of these fish traits spaces in the Supplementary Information, and we derived the conclusion you mentioned based on this Table S5.

According to your comments, we realized that interpreting the “range” and “area” of these traits spaces might, to some extent, be biased by a few “outlier” species. To improve the accuracy of interpretation, we have adopted your suggestion by focusing more on the “mean” of these traits. Plus, we double-checked the envelop shape and the distribution of fish species in the figure and removed those less convincing conclusions. We have updated this interpretation accordingly in the main text.

Specifically, we have changed to “Fish species associated with rainfed cropland had a relatively high mean and range of high body elongation (BEI); irrigated cropland-associated species showed relatively high maximum body length (MBL) and oral gape position (OGp), relative maxillary length (RMI), pertaining to overall high trophic levels which coincided with high nutrient loadings from surroundings; forest-associated species had comparatively small MBL and caudal peduncle throttling (CPT), suggesting small body size yet agility in swimming; urban-associated species had a high range of MBL, high value of body elongation (BEI) and CPT, and a large trait envelope area, indicating various strategies to adapt to high environmental and hydrological disturbances.”

DISCUSSION

The curious increase in species richness in impacted areas was carefully interpreted, and the paper correctly acknowledged past investigations. Yet, I would avoid some common-sense expectations like “it did not suggest that cropland expansion would always benefit fish biodiversity protection” (ln. 312). Obviously, increases in richness must not be associated per se with biodiversity conservation – context is required to interpret any relationship. Therefore, I suggest interpreting increase/decrease trends (or

positive/negative effects) in terms of biodiversity “change”. It is a more neutral, less prone to cause confusion or misinterpretation.

RESPONSE:

We have deleted the first part of this sentence and modified some sentences. Thanks for your suggestion.

Another relevant point is fauna homogenization due to habitat simplification. The loss of habitats and spatial heterogeneity is severe and may induce local gains but regional losses. The topic was barely touched, and I think it could be better discussed in the context of the present investigation (expansion of croplands).

RESPONSE:

This is also a very interesting point. We had examined the LULC change during 1992—2016 in this catchment yet found a slight change (~3% increase) in cropland area which was mainly distributed far from the river channels. And according to the fish species richness projections, these changes were rather small with only remarkable increase observed at the upper reach of Ping and Nan rivers. We would love to discuss this topic based on present investigation; however, the data itself has limited this discussion.

An important open question about stream integrity is the relative importance of the riparian buffer versus the catchment. The riparian buffer is well recognized, but distant effects from land use changes has not been evoked (or demonstrated) as a main driver of local changes. The present study is focused on this issue, but it was not emphasized. This topic is important from a management point of view, because legislation usually address each scale differently. In Brazil, for example, the protection of the riparian buffer is mandatory, but legislation is more permissible about the removal of vegetation in the catchment beyond the river channel. This scenario is relevant.

RESPONSE:

Thanks for your suggestion and this is a good point especially for linking our research to practical conservation planning. Indeed, we do agree the importance of assessing the influential distance of terrestrial LULC effects as was pointed by you. With your suggestion, we have explicitly added sentences about how our results could potentially contribute to conservation planning in the last paragraph of the Discussion section.

A main stressor is river fragmentation. Although this stressor was commented in the Introduction section, the Discussion ignored it. Consider addressing its potential use in spatial modeling coupled with LULC analysis.

RESPONSE:

We agree that river fragmentation is one of the major drivers of fish diversity loss. In the Chao Phraya River Catchment, there are two large dams, namely Bhumipol Dam (between site #2 and #3) and Sirikit Dam (between site #26, 27 and #28, 29; site #28 is not in the main channel yet near the river dam), located in the major river channels. According to the field sampling result, the fish species richness of site #2, 3, 26, 27, 28, 29 was 42, 30, 45, 21, 13, 33, respectively. So, sites (site #2 and #27) in the upper stream part of the river dam were more diverse in fish species than sites downstream, simply because there were large areas of croplands nearby. Therefore, we did not observe significant river fragmentation effect on the fish species richness pattern in this study catchment, so unfortunately unable to draw any solid conclusion pertaining to this topic. However, we are now conducting a river dam effect analysis using a global database and will especially assess the river dam effect on fish species distributions. This will be our next steps.

Thank you again for your valuable comments which helped us to improve the clarity and readability of this manuscript!

Reviewer #3 (Remarks to the Author):

This manuscript analyses the relationship between Land Use-Land Cover (LULC) and freshwater fish species richness and composition across a large river catchment from Thailand using eDNA samples to detect species presence and estimate richness values over the catchment. Although the manuscript is very well written and follows a clear and logical line and framework, I found quite difficult to understand in depth what the analyses are offering and what is the main message and value of the outputs found. The authors use a very complex methodology, which I acknowledge that I am not familiar with, but seems anyway too much in face of the general target, which is to relate two “simple” parameters, richness and LULC. Overall, their seems to be great creativity and novelty in the methodology, however I see no great novelty in the aims and results, that would justify publishing in *Communications Biology*, although that is an editorial decision, not a reviewer one. May be the manuscript would better fit in for instance *Ecological Indicators*? There is clearly an enormous analytical work in this manuscript, and sound results, which deserves being published for sure. I would suggest the authors to either simplify or explain why such complexity is needed to analyze their data and also make a more clear point on what are the main findings and their importance.

These general concerns being said, I still have one methodological comment related to the number of species detected. With eDNA sampling sites covering most of the catchment, authors detected around 100 species, while the freshwater fish richness in Thailand is estimated to be between 1000 and 1300, with circa 900 already described species, from which around a half may be present in the Chao Phraya basin. I would at least expect that the manuscript discuss this and explain why it’s not an issue, or how it could affect the results.

Another point to address or clarify is about the resulting patterns, namely a negative relationship of richness with forest and a positive one with rainfed cropland. When looking at the map, forests are placed in upstream areas and cropland downstream. Even if river discharge is accounted for in the analyses, could this result be related to the well-known increase in fish species richness along the longitudinal gradient of rivers?

RESPONSE:

Thank you for taking your time to read and comment on our manuscript. We respect and value your comments on methodological aspects. However, we respectfully disagree regarding the statements about novelty and relevance of our study, as we do not know of any other study that has combined eDNA and remote sensing-based data set to assess LULC effects on large fish communities, also demonstrating its potential for forecasting. If there are such studies, we are happy to be specifically hear about them. We are also convinced—alongside with the other two reviewers’ comments—that our study is highly relevant, especially in times of global biodiversity losses particularly pronounced in (sub)tropical streams.

Firstly, we would like to highlight the main message and the value of this study. River systems are suffering from severe biodiversity loss worldwide due to intense human impacts including land use change, direct exploitation, pollution, etc., as indicated by the global IPBES report. While this is generally acknowledged by scientists and stakeholders (and adequately referenced in our manuscript), neither an

explicit spatial scale nor a direct assessment of associations between aquatic and terrestrial systems has been systematically established. Importantly, there is a global consensus to bend this curve of decline in these ecosystems. Among all the human impact factors, LULC change has the strongest effect on freshwater ecosystems, as reported by IPBES Report. Although LULC effects on freshwater biodiversity has been extensively studied, there is still a lack of an effective tool to accurately assess human impact and, most importantly, to make quantitative predictions for the future. In riverine ecosystems, fish communities are the major and/or one of the most important components, playing an essential role in maintaining structure and diversity of riverine ecological communities. Consequently, our research successfully addressed the key and pressing issue of concern to biodiversity scientists and stakeholders. For these reasons, as also indicated by reviewers #1 & 2, we are confident that this study and the modeling framework substantially advance research in aquatic biodiversity, conservation design, and cross-ecosystem studies, making it especially suitable for a broad and general readership.

Second, we would like to justify the simplicity of our modeling approach. Actually, we had tried simple and straightforward methods in our preliminary analysis but it did not work well. At first, we performed correlation analyses of fish diversity versus area ratios of each LULC type across upstream flow distances (Figure R3). However, these correlation analyses could only indicate vague increasing (positive) or decreasing (negative) trends on fish species richness, without the ability to estimate the relative contribution and the influential distance, let alone forecasting fish diversity patterns across LULC change scenarios. This simple and straightforward approach, like many previous studies about fish-LULC relationship, did not provide sufficient and convincing spatial information to the core question.

Figure R3 Correlation of fish diversity versus area ratios of each LULC type across upstream flow distances. This was our very preliminary analysis about fish-LULC association. We adopted the original ESA CCI LC types (12 classes in this study) and directly conducted correlation analyses. Uncertainty was estimated and presented by using a bootstrap sampling framework. Red and green symbols represent negative and positive correlations, respectively.

The second attempt was to model this spatial association using machine learning (ML) methods. We soon gave up because 1) ML methods were like black boxes and did not provide a clear mechanism

behind, and 2) the number of sampling sites were somehow limited for large ML model training. Consequently, we decided to develop a new modeling approach in a quantitative and statistical manner.

This FishDiv-LULC modeling framework, essentially a classical spatial statistical approach, was inspired by a scientist in space physics who suggested to adopt a mechanical way to build up a model and then to use the observational records for estimation. In this spirit, we developed this spatially explicit modeling framework in a format that specifically makes all parameters meaningful. Although the model might look a bit complicated at the first glance, all the mechanical process behind is relatively simple and intuitive. Briefly, suppose we have fish sampling sites in the river and we have the site catchments. We assume the observed biodiversity value is associated with the sum of contributions from all LULC types times a distance decay framework, then plus a baseline estimation (see Figure R4 for a schematic diagram). All the parameters including LULC effect values and spatial scale can be effectively estimated using the classical statistical optimization method (e.g., when you use statistical function in R).

Figure R4 Schematic diagram of the FishDiv-LULC model.

For example, terrestrial LULC effect values have specific unit in num. species/km² depicting the magnitude of influence and the effective distance has unit in km suggesting the maximal upstream range of such spatial association. Therefore, with only a few parameters (spatial scale, magnitudes, baseline estimation coefficients), this FishDiv-LULC model showed a very good fitting result (adj.R²=0.587) and successfully captured the spatial variation of fish species richness pattern in such a large-scale and heterogenous catchment. Moreover, the model can also be used to map species habitat given terrestrial LULC and river channels properties.

Given all the above reasons, we are confident that the FishDiv-LULC model is the most concise and suitable model to evaluate such fish-LULC associations. As already explained in the Discussion section and indicated by reviewers #1 & 2, our approach would be very useful for conservation design and management and policy making.

With regard to the fish species pool, we definitely agree that Thailand is a hotspot for freshwater fish species. Before we conducted this research, we had checked the number of fish species in the Chao Phraya River and we found 329 fish species were reported on the FishBase database. Moreover, according to the report from the Freshwater Ecoregions of the World (FEOW), there are around 280 fish

species in the river. All these numbers include fish species that live not only in the mainstem but also in the small headwaters where we did not take eDNA samples. Seeing these, we thought that our eDNA sampling was robust enough to uncover the fish diversity pattern especially in the mainstem of the Chao Phraya River system.

Here we highlight again our best efforts to improve the detection ability using eDNA sampling (already mentioned in the response to reviewer #1 & 2). Basically, we took six samples (two samples in the middle of channel, two on the right and left side, respectively) with a relatively large volume (600 mL for each replicate). In the eDNA sequencing, we adopted two primers, i.e., Kelly and MiFish, built a customized fish eDNA reference database, and merged the results by taking the higher read counts for each fish taxon after removing very rare OTUs from each dataset (a very standard and robust way of eDNA processing). Our eDNA sampling and processing procedures were significantly better than many of the previous eDNA fish studies. In summary, it was unavoidable to have mismatch between sampled species pool and the whole species pool in that river for both traditional and eDNA sampling (traditional methods were even more difficult especially in large river channels); however, we had achieved our best for this sampling campaign and obtained good and robust results.

Lastly, we have found that the longitudinal gradient of rivers was not the driver of the gradient of fish species richness pattern especially in the mountainous region. In the response to reviewer #1, we have presented Figure R01 to further corroborate terrestrial LULC effects. Basically, we classified all the mountainous sites into “sites near cropland” and “sites away from cropland” according to cropland locations (whether spatially close to river channels) and percentages in area within a 6 km flow distance catchment upstream. Sites #1, 2, 4, 10, 12, 14, 16, 17, 18, 20, 25, and 26 were identified as “sites near cropland”. We found the mean of observed fish species richness for the two groups and found the average fish species richness was remarkably higher at “sites near cropland” (mean = 34.9) than “sites away from cropland” (mean = 20.9). In addition, we observed very high fish species richness values at site #1, 2, 20, 25, 26 (41, 42, 41, 42, 45, respectively; sites close to large cropland areas) which were at the same level as sites at the downstream channels. Therefore, it would be untenable to ascribe such a spatial variation mainly to longitudinal gradient of rivers. We do agree such a gradient plays a role so had included a river discharge as a proxy for baseline estimation. This factor proved to be already enough according to the analysis result.

To ensure that our aims and the essence of our research are clear, we have now elucidated these points in the above paragraphs. We are confident that this gives a better understanding and recognition of the uniqueness and importance of our study. Thanks again for your comments.

- Blackman, R. C., Osathanunkul, M., Brantschen, J., Di Muri, C., Harper, L. R., Mächler, E., Hänfling, B., & Altermatt, F. (2021). Mapping biodiversity hotspots of fish communities in subtropical streams through environmental DNA. *Scientific Reports*, 11(1), 1-11.
- Brantschen, J., & Altermatt, F. (2023). Contrasting strengths of eDNA and electrofishing compared to historic records for assessing fish community diversity and composition. *Canadian Journal of Fisheries and Aquatic Sciences*, 81(2), 178-189.
- Carraro, L., & Altermatt, F. (2022). Optimal Channel Networks accurately model ecologically-relevant geomorphological features of branching river networks. *Communications Earth & Environment*, 3(1), 125.
- Deiner, K., & Altermatt, F. (2014). Transport Distance of Invertebrate Environmental DNA in a Natural River. *PLOS ONE*, 9(2), e88786.
- Egeter, B., Peixoto, S., Brito, J. C., Jarman, S., Puppo, P., & Velo-Antón, G. (2018). Challenges for assessing vertebrate diversity in turbid Saharan water-bodies using environmental DNA. *Trends in DNA Barcoding and Metabarcoding*, 01(01), 807-814.
- Keck, F., Blackman, R. C., Bossart, R., Brantschen, J., Couton, M., Hürlemann, S., Kirschner, D., Locher, N., Zhang, H., & Altermatt, F. (2022). Meta-analysis shows both congruence and complementarity of DNA and eDNA metabarcoding to traditional methods for biological community assessment. *Molecular Ecology*, 31(6), 1820-1835.
- Nakagawa, H., Yamamoto, S., Sato, Y., Sado, T., Minamoto, T., & Miya, M. (2018). Comparing local- and regional-scale estimations of the diversity of stream fish using eDNA metabarcoding and conventional observation methods. *Freshwater Biology*, 63(6), 569-580.
- Penaluna, B. E., Allen, J. M., Arismendi, I., Levi, T., Garcia, T. S., & Walter, J. K. (2021). Better boundaries: identifying the upper extent of fish distributions in forested streams using eDNA and electrofishing. *Ecosphere*, 12(1), e03332.
- Pont, D., Rocle, M., Valentini, A., Civade, R., Jean, P., Maire, A., Roset, N., Schabuss, M., Zornig, H., & Dejean, T. (2018). Environmental DNA reveals quantitative patterns of fish biodiversity in large rivers despite its downstream transportation. *Scientific Reports*, 8(1), 10361.
- Tanaka, W., Wattanasiriserekul, R., Tomiyama, Y., Yamasita, T., Phinrub, W., Chamnivikaipong, T., Suvarnaraksha, A. and Shimatani, Y. (2015). Influence of Floodplain Area on Fish Species Richness in Waterbodies of the Chao Phraya River Basin, Thailand. *Open Journal of Ecology*, 5, 434-451.
- Williams, K. E., Huyvaert, K. P., & Piaggio, A. J. (2017). Clearing muddied waters: Capture of environmental DNA from turbid waters. *PLOS ONE*, 12(7), e0179282.
- Yang, H., Du, H., Qi, H., Yu, L., Hou, X., Zhang, H., Li, J., Wu, J., Wang, C., Zhou, Q., & Wei, Q. (2021). Effectiveness assessment of using riverine water eDNA to simultaneously monitor the riverine and riparian biodiversity information. *Scientific Reports*, 11(1), 24241.

Response to reviewers

Reviewer #1 (Remarks to the Author):

Most of my concerns were addressed or explained by the authors in this reviewed version.

The main limitation of the study of using 300-m resolution LULC images are difficult or even impossible to be resolved with the current data available worldwide, as claimed by the authors. I do think, however, the overall patterns of the results are still valid and consistent, although the exact effect size and distance-weighted LCLU values could change if a more refined resolution were available. On the other hand, I think authors could discuss this limitation briefly somewhere in the Discussion or in the Methods.

RESPONSE:

Thank you very much again for your detailed and constructive comments, as they were indeed very helpful in improving our manuscript. The choice of 300 m resolution was a compromise between high level of LULC details and globally available datasets where regions of the world that are well-resolved (Europe, North America, etc.) and beyond. Importantly, however, a coarser or finer resolution itself is not per se impeding our analysis; it may simply lead to a relatively low granularity of interpretation. Followed again by your suggestions, we have added some sentences about the relevance of spatial resolution of the input data in the Materials and Methods section (after the “River channel and catchment data”).

I have some addition comments, however:

Discussion

L258: the amount of forest and/or cropland and urban land use are usually negatively correlated. So I recommend being careful in saying that forest has a negative effect of richness. This is most likely a positive effect of cropland related to the factors already addressed by the authors, rather than a negative effect of forest. Please, review/rewrite such a statement here and elsewhere in the results and discussion.

RESPONSE:

This is a fair point. We do agree that the “negative” word might be misinterpreted by general readers. For this reason why we always emphasized “relative negative” in the manuscript, yet this may not in all places be completely clear. Thus, we have rewritten the sentences in the Result and Discussion sections and especially emphasized that they were the difference in levels of contributions. We especially used “relative negative values” instead of “relative negative effects” for forest and urban area.

301-322: The authors decided to maintain the trait-based approach in the manuscript. However, now they are presenting the results for this part in the Discussion. I do think since they decided to keep this approach they should present these results in the Results section and better use this paragraph to explain the meaning of such results.

RESPONSE:

Thanks for your further suggestions. In the previous revision, we actually had moved this

figure/paragraph to the Discussion section because of the suggestion from the second reviewer. We agree that this is an essential part of the result that helps a general reader to understand the possible mechanism behind, although this fish trait analyses could also be suitable for a paragraph in the Discussion section in terms of the logic and story line perspective. We have ultimately decided to move the fish trait analyses back to the Results section, as suggested by you, to alleviate difficulties in understanding. Given that we have here two different suggestions from reviewers (one preferring it to be in Method section, the other in Discussion section), we respectfully ask the editor to help clarify what would work best, or if it is fine to keep it in the Result Section (as it is now, and where it is likely classically expected to be as a result).

323-342: I notice the authors included a discussion about homogenization as recommended by one reviewer. I recommend, however, to be careful when talking about homogenization, because some recent papers have been showing that human modifications can also cause differentiation of communities, which can be as bad as homogenization. The spatial and temporal scale addressed in the study and in the discussion will have an important implication on whether you will see homogenization or differentiation of communities. So please make it explicit what is that spatial (within river? between rivers? in the whole catchment?) and temporal scale (between years? or along the time?) you are discussing in this paragraph and also what is the diversity metric (taxonomic vs. functional vs. genetic? Species level? Family level?) you are talking about.

RESPONSE:

We do agree the possible differentiation of aquatic communities caused by human modifications and your suggestion of being careful when interpreting homogenization, and this effect has actually been found by a recent study by Keck et al. 2025 in Nature (showing that human impacts can cause either homogenization or differentiation). Accordingly, we have rewritten those relevant sentences and changed “homogenization” to “decrease in uniqueness”.

With regard to spatial scales, we always explicitly referred to as “the spatial range of terrestrial LULC effects”, which means the spatial scale of land-water linkage within the whole catchment (only considering terrestrial pixels as we removed water pixels in the modeling). We have noted that in the last paragraph of Discussion section, the use of “spatial scale” caused ambiguity, so have already changed the “spatial scale” to “spatial distance” for consistency.

In the last two paragraphs, we discussed the seasonal change of terrestrial LULC effects in a very generic manner as they are the future steps for following research. Therefore, we could potentially consider all levels of diversity including taxonomic, functional, and genetic as mentioned by you. Plus, with eDNA samples across seasons and years, it is possible to assess changes in the strength of overall fish-LULC linkages and variations in terrestrial LULC effects. This could add a new dimension for future research.

Thank you very much again for your constructive, supportive, and very careful review. We appreciate a lot your contributions and efforts in helping us improving both the robustness and the clarity of this work!

Reviewer #2 (Remarks to the Author):

Dear Authors

Thank you for revising the manuscript and addressing reviewers' comments. The original submission was already very well-done, and the revised version is even more robust and clear.

I have no further comment to provide. Congratulations for this project, it is a great contribution to the field.

I recommend publication.

Regards

Fernando Pelicice

RESPONSE:

Thank you very much again for providing us constructive comments and suggestions which helped us to greatly improve the clarity of the manuscript. As pointed out by you, we hope the FishDiv-LULC model and this analysis will become a standard tool to assess terrestrial LULC effects on the distribution of fish and other aquatic taxa, and will help stakeholders better design the conservation area for aquatic biodiversity. We thank you again for highlighting the importance of our work!